# Spatial Congruence Analysis (SCAN): A method for detecting biogeographical patterns based on species range congruences

**Cassiano A. F. R. Gatto**[1]*, **Mario Cohn-Haft**[2]

**1** Pós Graduação em Ecologia—PPG-ECO, Instituto Nacional de Pesquisas da Amazônia—INPA, Manaus, Brazil, **2** Coleção de Aves, Coordenação de Pesquisas em Biodiversidade, Instituto Nacional de Pesquisas da Amazônia–INPA, Manaus, Brazil

\* cassianogatto@gmail.com

**Data Availability Statement:** Data on the simulated gradient analyzed is within the paper and its Supporting Information files. The range polygons used here are a subset of the Bird

## Abstract

Species with congruent geographical distributions, potentially caused by common historical and ecological spatial processes, constitute biogeographical units called chorotypes. Nevertheless, the degree of spatial range congruence characterizing these groups of species is rarely used as an explicit parameter. Methods conceived for the identification of patterns of shared ranges often suffer from scale bias associated with the use of grids, or the incapacity to describe the full complexity of patterns, from core areas of high spatial congruence, to long gradients of range distributions expanding from these core areas. Here, we propose a simple analytical method, Spatial Congruence Analysis (SCAN), which identifies chorotypes by mapping direct and indirect spatial relationships among species. Assessments are made under a referential value of congruence as an explicit numerical parameter. A one-layered network connects species (vertices) using pairwise spatial congruence estimates (edges). This network is then analyzed for each species, separately, by an algorithm which searches for spatial relationships to the reference species. The method was applied to two datasets: a simulated gradient of ranges and real distributions of birds. The simulated dataset showed that SCAN can describe gradients of distribution with a high level of detail. The bird dataset showed that only a small portion of range overlaps is biogeographically meaningful, and that there is a large variation in types of patterns that can be found with real distributions. Species analyzed separately may converge on similar or identical groups, may be nested in larger chorotypes, or may even generate overlapped patterns with no species in common. Chorotypes can vary from simple ones, composed by few highly congruent species, to complex, with numerous alternative component species and spatial configurations, which offer insights about possible processes driving these patterns in distinct degrees of spatial congruence. Metrics such as congruence, depth, richness, and ratio between common and total areas can be used to describe chorotypes in detail, allowing comparisons between patterns across regions and taxa.

Species Distribution Maps of the World V6.0 (BirdLife & HBW, 2016), publicly available at http://datazone.birdlife.org/species/requestdis.

**Funding:** CAFRG received a Doctoral Fellowship from the Conselho Nacional de Desenvolvimento Científico e Tecnológico - CNPq (142234/2015-0) and a sandwich grant from Coordenação de Apoio à Formação de Pessoal de Nível Superior - CAPES/PDSE (88881.188857/2018-01). The funders had no role in study design, data collection and analysis, decision to publish, or preparation of the manuscript.

**Competing interests:** The authors have declared that no competing interests exist.

# Introduction

"If there is any basic unit of biogeography, it is the geographic range of a species."—Brown, Stevens & Kaufman [1].

"[spatial] congruence [. . .] should be optimized, while realizing that this criterion will most likely never be fully met"—HP Linder [2].

Species ranges are determined by physical and biological processes at multiple scales of space and time [1–3]. Although the exact processes and timing are likely to be unique in each case, it is relatively common that several species share the same or similar global distributions [4, 5]. This intriguing phenomenon of shared distributions is an important building block in biogeography, making it possible to infer the action of shared causes [6, 7]. Thus, congruent ranges have long been a major source of insights for naturalists at the foundations of ecology and evolution [8–11].

Recognition of congruent distributions, however, has traditionally been a relatively subjective process with some unexpressed premises. When species show very similar distributions, it is natural to assume that strong ecological pressures play an important role in this congruence. For example, a bird species whose distribution coincides closely with that of a particular tree (e.g., *Picoides tridactyla* and *P. dorsalis* three-toed woodpeckers and *Picea* spp. spruce trees [12]) is readily interpreted as being dependent on the presence of those trees. On the other hand, coincident distributions of dozens of unrelated bird species in discrete parts of the Amazon basin are assumed to reflect past vicariant events that led to differentiation and geographic limitation in numerous groups simultaneously [13, 14]. In these latter cases, less precise congruence is implicitly acceptable because each species is likely to have adapted to modern environments in unique and slightly distinct ways. Degree of congruence, then, is an important variable to quantify and, ideally, use as an explicit parameter in any study of patterns based on common geographic distributions [2].

Patterns derived from congruences in geographical distributions can be analyzed in two epistemologically distinct ways [15]. The regionalization approach uses species' ranges to classify areas, often assuming these areas have unique histories of species diversification [16]. Conversely, the chorological (areographical) approach focuses on the identification of chorotypes, that is groups of species sharing similar geographical ranges, without *a priori* assumptions about tempo or mode of diversification [15, 17].

Methods to identify patterns of shared ranges have been adapted mostly from ecological analyses (e.g., [18, 19]), in which grid-cells dividing the studied area replace the role of sample units (e.g., collecting stations). The use of clustering and grid-based bipartite network approaches (e.g., one layer for species, another for grid-cells [20]) has numerous computational advantages. However, the grid-cells themselves become the information-bearing units, which may lead to scale and range distortions [21, 22]. In addition, studies inferring historical and ecological processes driving patterns of shared ranges may produce contrasting results according to the specific clustering method used (e.g., [18]), or overlook biogeographically meaningful patterns, such as independent overlapped spatial associations [23], or gradients of ranges of species dispersing away from a common nuclear area [24, 25].

Among numerous methods designed to search for species spatial relationships [2, 26–30], including improvements to standard regionalization protocols [31–35], some innovative strategies improving the identification of chorological units stand out. The *a posteriori* use of fuzzy-logic over chorotypes identified by agglomerative clusters (e.g., UPGMA) allows the classification of species according to grades of membership to these chorotypes and,

consequently, the detection of gradients and overlapped patterns [17, 36, 37]. In addition, some methods originally conceived as regionalization tools present original ideas improving the detection of shared patterns that can be incorporated by chorological studies. Endemicity Analysis [28, 38, 39], for example, identifies areas matching the distribution of as many congruent species as possible based on an explicitly defined criterion of "endemicity" (analogous to spatial congruence). Infomap Bioregion, a method based on random walks across a bipartite network, introduces the use of higher-order spatial relationships, greatly extending the (limited) capacity of other methods in detecting and describing complex gradients of species distribution [27, 40], appropriately described only through turnover methods (e.g., [32, 41]).

In this paper, we describe a new method for generating chorotypes, Spatial Congruence Analysis (SCAN). SCAN quantifies range congruences between species without assumptions related to processes or historical homology [15, 25], while explicitly exploring the role of more strict or relaxed congruence limits in pattern recognition. By allowing different degrees of congruence and indirect comparisons (see Methods) in a network of spatial relationships, this approach should recognize gradients of species ranges without confounding independent chorotypes. SCAN can use polygons (shape files) of species range maps, and by using entire ranges as the unit of comparison, rather than grid cells, problems of scale distortion are minimized. First, we exemplify SCAN's behavior and specific metrics with a hypothetical dataset and then further illustrate pros and cons of the approach with real bird species ranges in the rich and biogeographically complex South American avifauna. SCAN's mathematically simple and intuitive approach of direct range map comparisons offers an alternative to other available methods, using different computational or conceptual bases, and provides new parameters and metrics that can be useful in interpreting biogeographical patterns.

## Material and methods

### SCAN overview

SCAN is based on a script written in R environment [42], which relied mostly on *sf* package [43] for spatial tasks (S1 Script). In a nutshell, it uses a quantitative measure of spatial congruence (see below) to compare polygons of species ranges. Comparison of a chosen initial species with all others yields a set of species whose ranges are spatially connected at a given threshold of congruence. Subsequently, each of those already grouped species is also compared to all others, sometimes accruing more species to the list. The process is repeated until no more ranges are found to be congruent at the threshold being evaluated, thus forming a "closed list", which we refer to as a "partial chorotype", because it was created at a particular level of congruence.

The process begins at the highest congruence threshold (e.g. 100%), saves the results of the iterative comparisons (if any congruence is found and if the group closes), and then reinitiates using the same initial species for the next round with a lower threshold (e.g. 99%). Each time a new partial chorotype is detected (i.e., that level of congruence leads to closure), this list of species, sometimes more inclusive, is saved. This process continues anew at each progressively lower congruence threshold until the comparison no longer leads to closure. The set of all partial chorotypes derived from a particular reference species, at all congruence levels, corresponds to the "chorotype", comprising all potential taxonomic and spatial variations identified at distinct threshold levels.

### Congruence index

Determining spatial congruence is not a trivial problem because ranges may differ in position, area, and shape. Two ranges of equal area and shape may vary in the amount of overlap, just as

two ranges of equal shape and central position may differ in size, and yet areas of the same size and position may overlap only slightly if their shapes are very different ([Fig 1A]). To distill these differences into a single index, spatial congruence between two species can be calculated by the product of area of overlap weighted by the relative area of each (see equation below). This generic spatial index is analogous to the Jaccard index and was proposed in the "Goodness of Fit" method to compare maps of Hargrove et al. [44] and is hereafter referred to as the

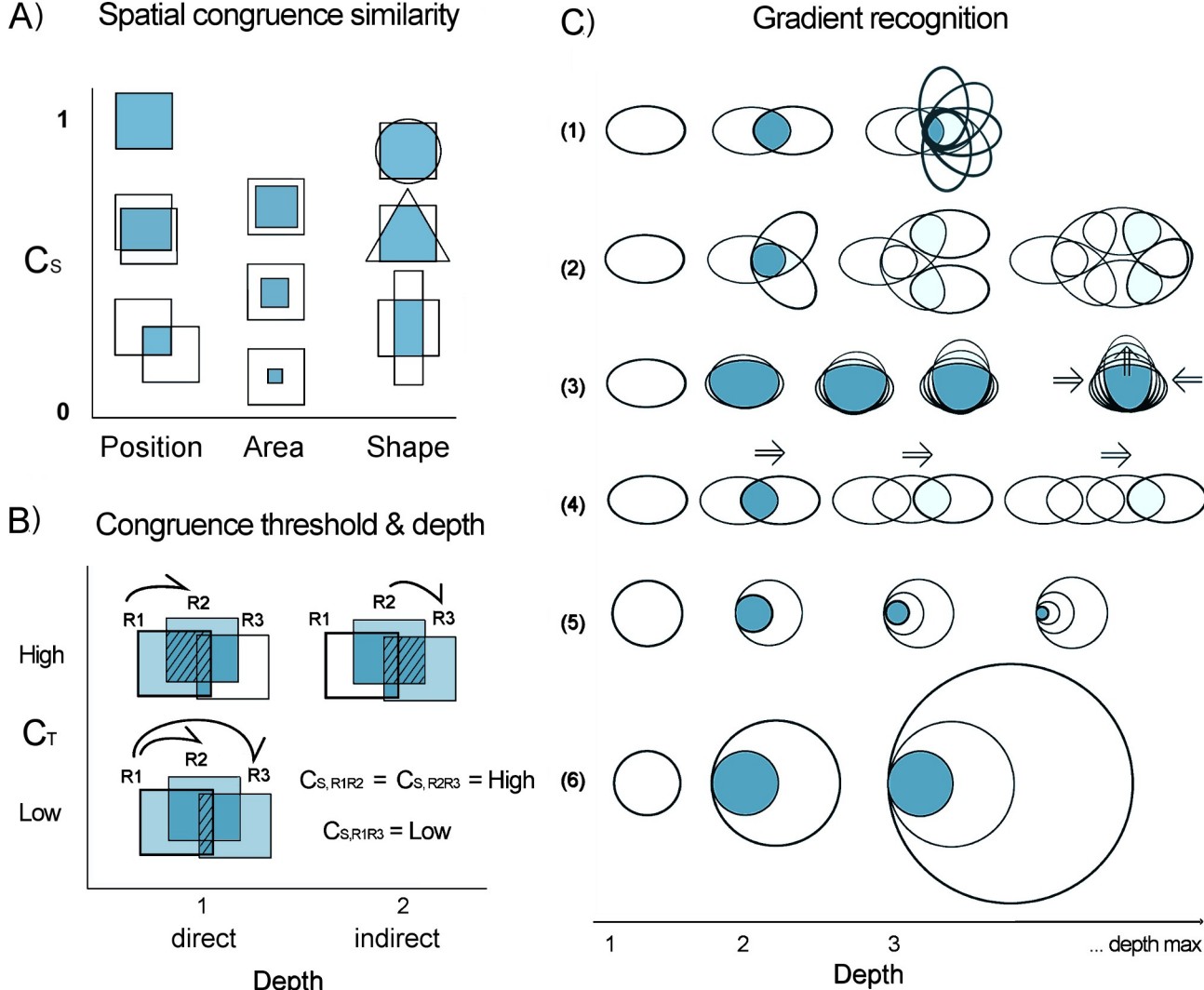

**Fig 1. Schematic representation of the theoretical behavior of spatial congruence with respect to range similarity, threshold values, depth of direct and indirect congruences, and gradient recognition.** All figures are spatial representations; darker tones indicate overlapped (shared) ranges. (A) The $C_S$ index (y-axis) compares position, extent, and shape between two species ranges. For identical size and shape, position determines the amount of overlap; when shape and position (e.g., centroids) are the same, one area will reside entirely in the other and difference in area alone will determine their congruence; and, finally, if area and position are the same, shape will be determinant. (B) The congruence threshold ($C_T$) is the reference for spatial relationships. Range R1 is directly congruent with R2 when their $C_S$ (calculated from their area of overlap–hashed area) is greater than or equal to $C_T$. Indirect relationships occur when ranges are linked by congruence in a concatenated chain of $C_S \geq C_T$. R1 is indirectly congruent with R3 because R2 is as congruent (same $C_S$) to R3 as it is to R1. At relaxed threshold requirements (i.e., $C_T < C_{S, R1R3}$), R2 and R3 are both directly related to R1. (C) Indirect links allow the recognition of many types of syndromes of shared distributions potentially associated with distinct historical or ecological range drivers. Links may, at least theoretically, lead to nuclear regions of congruence (example1), congruence zones following patches of favorable habitat (e.g. mountain slopes, or riverine habitats; ex. 2), gradients of expansion or contraction (ex. 3), or linear gradients (ex. 4). Indirect congruence (depth) at very low congruence thresholds may also lead to scalar distortions (ex. 5,6).

"Spatial Congruence Index"–$C_S$, defined as follows:

$$C_{S,ab} = \left(\frac{O_{ab}}{A_a}\right) \times \left(\frac{O_{ab}}{A_b}\right)$$

where $C_S$ is the spatial congruence index, $O_{ab}$ is the area of overlap, and $A_a$ and $A_b$ are the areas of the ranges of species *a* and *b*, respectively. The index varies from zero (no congruence) to one (perfect congruence) and can also be expressed as a percentage. It allows the use of vector-based distributions (polygons), thus avoiding the scale-dependent shape distortion caused by the use of coarse grid-cells [29]; however, this index can also be used with grids. Furthermore, any other quantitative congruence index may be substituted in the method here proposed, if preferred or if required to compare other range representations, such as the probabilistic surfaces resulting from species distribution models.

### Direct versus indirect congruences

Spatial congruences, here, are always evaluated with respect to a specific congruence threshold ($C_T$). Two species are directly congruent when their calculated $C_S$ is greater than or equal to a particular $C_T$ (Fig 1B). Indirectly related species are described here as those linked through a chain of direct connections using a third linking species. For example, if species ranges *A* and *B* are directly congruent at a given $C_T$, and *B* and *C* are also directly congruent, then *A* and *C* are indirectly congruent at this $C_T$, using *B* as a link ($A \leftrightarrow B \leftrightarrow C$). These direct and indirect relationships are recovered and organized by the algorithm at every referential $C_T$ value. Thus, for a given reference species, the method begins setting the current $C_T$ and comparing the spatial distribution of the reference species to the range of all other species. At each $C_T$ analyzed, any species with $C_S \geq C_T$ to the reference is directly congruent. The next pass involves comparing each of these directly congruent species to all other species. Any additional species added in this second pass (and all subsequent passes) are indirectly congruent to the reference species at the same $C_T$. The number of passes required such that the species group closes (i.e., no additional congruent species are included) is equal to the length of the longest indirect chain and is called the "depth" of that species group (Fig 1B and 1C). Depth is a metric that emerges from the analyses; however, its exact biological significance and its computational utility are not clear, but may be subject to future study.

Including indirect congruences in the recognition of partial chorotypes has two distinct advantages. First, it allows the recognition of syndromes of shared distributions (Fig 1C). The set of species in a partial chorotype, at any given $C_T$, will be very similar to that formed by starting the analysis with other members of the group. Thus, this is a "natural" grouping. Second, indirect congruences allow the recognition of gradual relationships among species ranges (Fig 1C). Although it is conceivable that virtually all species be related to one another through such gradual indirect congruence, this is controlled by congruence threshold requirements. In practice, this rarely happens; rather, groups of species sharing biogeographical properties may close even at fairly low $C_T$ (see Results).

### Control parameters

All reference species have their $C_S$ calculated with respect to all species available in the whole pool of taxa. A list of species to be analyzed as references and maps for all taxa are the basic inputs required to run the algorithm (a tutorial is available at S2 Script). The most important pre-analysis settings are: 1) the maximum depth allowed; 2) maximum and minimum $C_T$ (highest and lowest congruence limits defining the range of thresholds to be analyzed); 3) $C_T$ resolution (interval size between each subsequent threshold analysis). Max depth and min $C_T$

are limiting parameters: at any threshold, if any one of these limiting values are reached, the analysis for the current reference species finishes. Empirical evidence collected from the bird dataset analysis (see S2 and S3 Figs) showed that relaxed settings, such as the default values, allow the recognition of most natural groupings derived from a particular reference species. The default values of max depth (7) and the default limiting range of congruence thresholds (1 max and 0.1 min), for example, are rarely attained. Partial lists usually close at depths less than 5, and do not reach congruence thresholds as low as 0.1 (10%). Similarly, the default $C_T$ resolution at 0.01 (1% increments) showed a good compromise between the level of detail in the description of compositional changes in partial chorotypes and computation time. More restrictive customized settings can be used in analyses targeting specific objectives (e.g., ignoring longer chains of indirect relationships, or focused on a specific congruence threshold range). In addition to these three numerical values, the user may stipulate a criterion of spatial overlap. If adopted (default), this criterion requires that a congruent range must include at least some area of overlap with all previously grouped species.

Once these parameters are chosen, the iterative procedure of evaluation of spatial relationships across congruence thresholds will be executed for all references, one by one. At any time, the analysis will stop at a given $C_T$ with a given reference species when a group closes, that is, the next depth of indirect comparisons adds no new species. That closed group is recorded and the comparisons begin again at the next lower $C_T$. The entire sequence of comparisons for a given reference species stops when groups no longer close before reaching the stipulated maximum depth or the minimum congruence threshold, or do not meet the spatial overlap criterion (if adopted). When the stopping point is reached for a given reference species, another species is run to completion, and so on until all reference species have been analyzed.

## Partial chorotypes, chorotypes, synonyms, and metrics

The partial chorotype will always be dependent on a specific, explicit congruence threshold and reference species, and will also be associated with a particular depth (see S1 Fig). The partial chorotype, thus, is a suite of species. The spatial area determined by their distributional ranges also reveals two important spatial characteristics: a "total area", which congregates the ranges of all grouped species, and a "common area", delimited by their spatial intersection. One simple metric, the "ratio between common and total areas" (intersection/union), ranges from 0 to 1 and addresses the trade-off between spatial comprehensiveness and cohesion (see Discussion).

SCAN differentiates those reference species giving rise to or composing partial chorotypes, ("informative species") from those not participating in any closed groups, at any congruence threshold ("non-informative species"). For any informative reference species, the set of all partial chorotypes over the range of congruence thresholds constitutes a chorotype. "Synonymous" chorotypes (or simply "synonyms") are those derived from distinct reference species that converge on the same group of taxa. Nested chorotypes are those composed by subsets of larger chorotypes. Parameters such as maximum and minimum $C_T$, depth, and derived metrics, such as the number of species, are important metrics that can be used to identify synonyms and compare partial chorotypes or whole chorotypes derived from distinct species or regions (S1 and S2 Tables). Although synonyms and nested chorotypes are overlapped by definition, spatial overlap may also occur between independent chorotypes, with no species in common (see Results). Here, the set of synonymous chorotypes generated by different reference taxa are represented by and named after the reference taxon with the highest mean congruence ($C_S$) with all grouped species (S3 Table).

## Applications of the method

To explore and illustrate this protocol we use two datasets, one hypothetical and the other of real bird species distributions. First, a hypothetical set of species ranges was used to evaluate the capacity of SCAN to detect gradients. In addition, the effects of maximum depth settings on pattern detection were explored (see S2 Script). The problem, proposed by Kreft and Jetz [45], is when two seemingly distinct sets of species (for example a northern and a southern group) contain a succession of species with ranges each slightly greater than the previous such that those with the most extensive ranges in each group actually overlap partially the widest-ranging species of the other group (Fig 2). Any biogeographical method would be seen to fail if

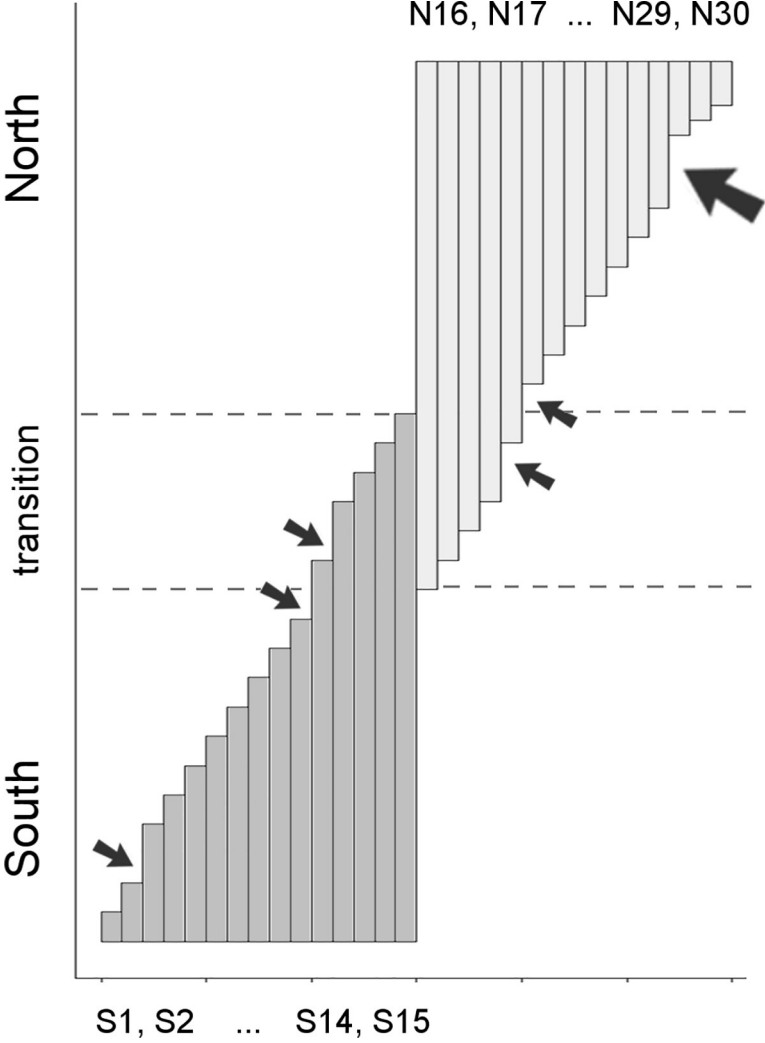

**Fig 2. Schematic representation of the hypothetical gradient of thirty species ranges proposed by Kreft & Jetz [45].** Two well delimited biogeographical centers of diversity, a northern (pale gray) and a southern (medium gray), have species extending their ranges through a zone of transition. Each vertical bar represents the latitudinal extent of the range of a particular species (named S1-15 and N16-30). The longitudinal ranges (bar widths and position) of all species are assumed to be identical, such that they all overlap spatially. The original symmetric scheme [45] included a few small exceptions (small arrows) to the otherwise uniformly graded range differences. To further examine the influence of non-uniform gradations on detection of congruence patterns, we reduced the range differences among N28-30 relative to the others, which adds a single slightly larger range discontinuity (large arrow).

it did not recover the two groups as separate units or, if it identified the transition zone between them as an independent unit, which is, obviously, a false conclusion.

For subsequent exploration of the method, a dataset with real species distributions from BirdLife & HBW [46] was used to examine the relationships between parameters such as $C_S$, $C_T$, and depth, and to illustrate the method with maps of real patterns detected. The set of taxa analyzed as reference species is here defined arbitrarily as those 1095 birds for which at least 50% of their distribution is within Amazonia [47]. To allow recognition of possible patterns that extrapolate the study area (Amazonia), those species' ranges were compared to a larger dataset containing all 3083 New World bird species whose ranges overlap the Amazon even only slightly. The complete ranges of all involved species were analyzed and, thus, the resulting chorotypes are global, rather than regionally delimited [4].

## Results

### Gradient simulation

Analysis of the simulated gradient resulted in a total of 23 unique partial chorotypes (S1 and S2 Tables; Fig 3). SCAN always correctly recognized north and south as distinct groups. The number of informative species detected varied according to max-depth settings (S2 Table). At the shallowest depth limit (maximum depth = 3), approximately half of the species gave rise to chorotypes. At the extremely relaxed max-depth = 10, all reference species expanded their indirect congruences to encompass either the entire southern or northern groups, but never included species from the opposite group if the "spatial overlap criterion" (see Methods) was implemented. Intermediate scenarios with alternative and plausible classification schemes were already achieved at depth limits of 5 and 7, but always nested within the overall north-

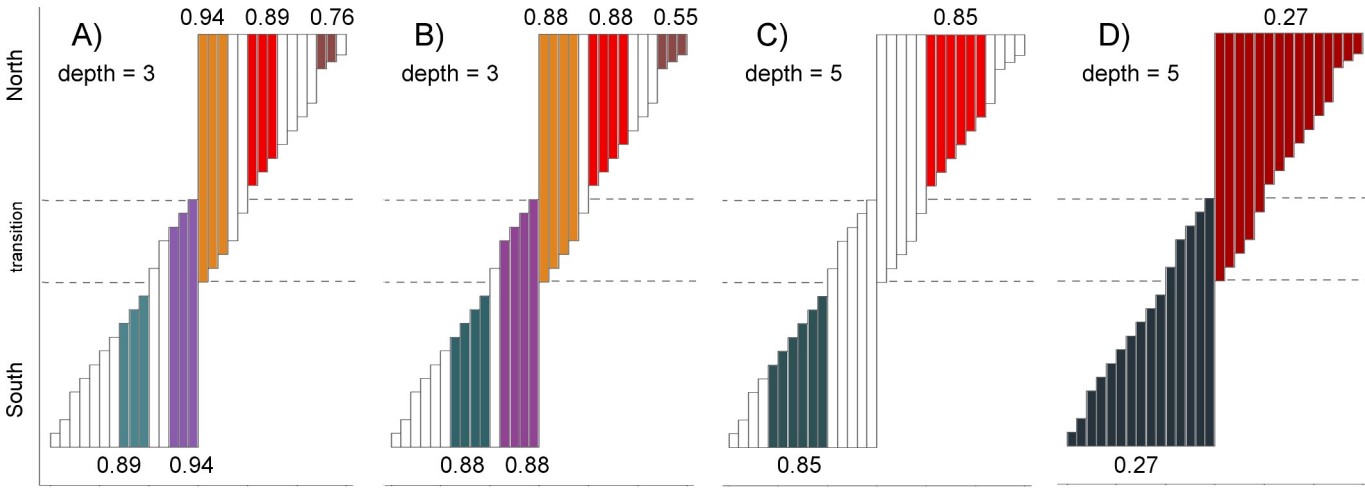

**Fig 3. Partial chorotypes (closed lists) detected in a simulated gradient of ranges.** Like colors indicate species ranges (bars) grouped into partial chorotypes recovered at particular congruence thresholds (minimum $C_T$ values are shown at the base and top of each closed group). White bars indicate "biogeographically uninformative" species that did not give rise to and were not included in any closed list, at each corresponding maximum depth setting. Four examples are shown (see S1 Table for full details). A) Small but highly congruent partial chorotypes (closed groups) were located next to zones of range discontinuity and favored larger ranges, with smaller proportional area differences. The extended discontinuity only at the northern extreme allowed the recognition of one pattern composed of small-range species. B) At lower congruence thresholds (e.g., from 0.94 to 0.88, or 0.89 to 0.88, or 0.76 to 0.55; compare Fig 3A and 3B), partial chorotypes included adjacent species with smaller ranges. C) Increasing depth (allowing more levels of indirect congruences) increased the size of partial chorotypes at relatively high congruence thresholds. D) At lower congruence thresholds, the entire northern and southern groups were recovered as a whole. Allowing for extensive indirect range comparisons (maximum depth of 10, not shown), most reference species led to convergence on a single chorotype. Note that all recovered partial chorotypes can be nested in others and there was no non-nested overlap nor recovery of the transition zone as an independent chorotype.

south distinction (Fig 3). These alternative configurations of partial chorotypes are ideal to show how $C_T$ relaxation allows the incorporation of slightly less-congruent species, either through direct or indirect connections.

### Bird dataset

"SCANning" the entire 1095-species Amazonian bird dataset resulted in a very large suite of chorotypes with interesting implications for the study of Amazonian biogeography. That subject goes well beyond the objectives of this paper and will be presented in detail separately. We examined the overall performance of the method running exploratory analyses in R to synthesize relationships between parameters and metrics. These analyses used relaxed (default) settings, which allowed recovery of most, if not all, potential species inter-relationships in pilot tests (see below). Specifically, the cases of *Icterus nigrogularis* (Yellow Oriole) and *Amazilia tobaci* (Copper-rumped Hummingbird), at the northern extreme of the Amazon (Fig 4), and a genus of hummingbirds (the "brilliants", *Heliodoxa* spp.), with a large variety of habitat associations and peculiar ranges (Fig 5), are sufficient to demonstrate here the main properties and characteristics of SCAN for real life distributions.

Roughly half of the bird species analyzed as references (558) were recognized as informative by the algorithm. These species gave rise to approximately 150 chorotypes, including synonyms and nested units, grouping anywhere from two to 99 species. Preliminary explorations showed that the algorithm chooses a very limited set of highly congruent range overlaps (S2 Fig). Chorotypes based on reference species with larger areas had higher $C_S$ means (i.e. the average $C_S$ calculated between the reference species and all its overlapping ranges). Range areas and $C_S$ means had no effect on the number of species (richness) in chorotypes, but relaxed congruences and depths led to patterns with greater species richness (S3 Fig). The full set of graphic preliminary explorations is presented as S2 and S3 Figs.

Chorotypes derived from this real-life avifauna were highly diverse in many aspects (S1 Table; Figs 4 and 5). The simplest patterns had only one partial chorotype with a fixed group of species, usually in a very limited range of threshold values, as shown by the lowland hummingbird *Heliodoxa aurescens* (Fig 5A). Alternatively, "shallow" patterns added species across a range of congruence thresholds but showed no indirect connections, grouping species only at the first depth level (Figs 4B, 5D and 5F). The depth 'dimension', as predicted, revealed gradients through indirectly related ranges. This potential is elegantly illustrated by the *Icterus nigrogulari*s chorotype, which, adding species by species in a series of partial chorotypes across progressively less restricted congruence thresholds, gradually extended its total area through the Guianan coast and eventually reached the central Amazon via the Amazon river channel, at low congruence thresholds (Fig 4A).

SCAN can recognize more than one chorotype centered on essentially the same general area, but with considerably different spatial limits and containing mutually exclusive sets of species, as demonstrated by *I. nigrogularis* and *A. tobaci* (Fig 4). Similarly, the *H. xanthogonys* pattern grouped highland birds of the Tepuis region that are completely overlapped (at this geographic macro-scale) with birds typically associated with lowland patterns (Fig 5C). Other *Heliodoxa* hummingbirds illustrate SCAN's accuracy recognizing patterns derived from species with small, somewhat linear ranges, sometimes at very low congruence thresholds, based on subtle similarities of range position and shape, even grouping species with similar disjunct distributions (Fig 5D and 5F).

### Discussion

SCAN proposes a comprehensive description of biogeographical scenarios based on chorotypes derived from the whole web of spatial interrelationships of species in a biota. Explicit

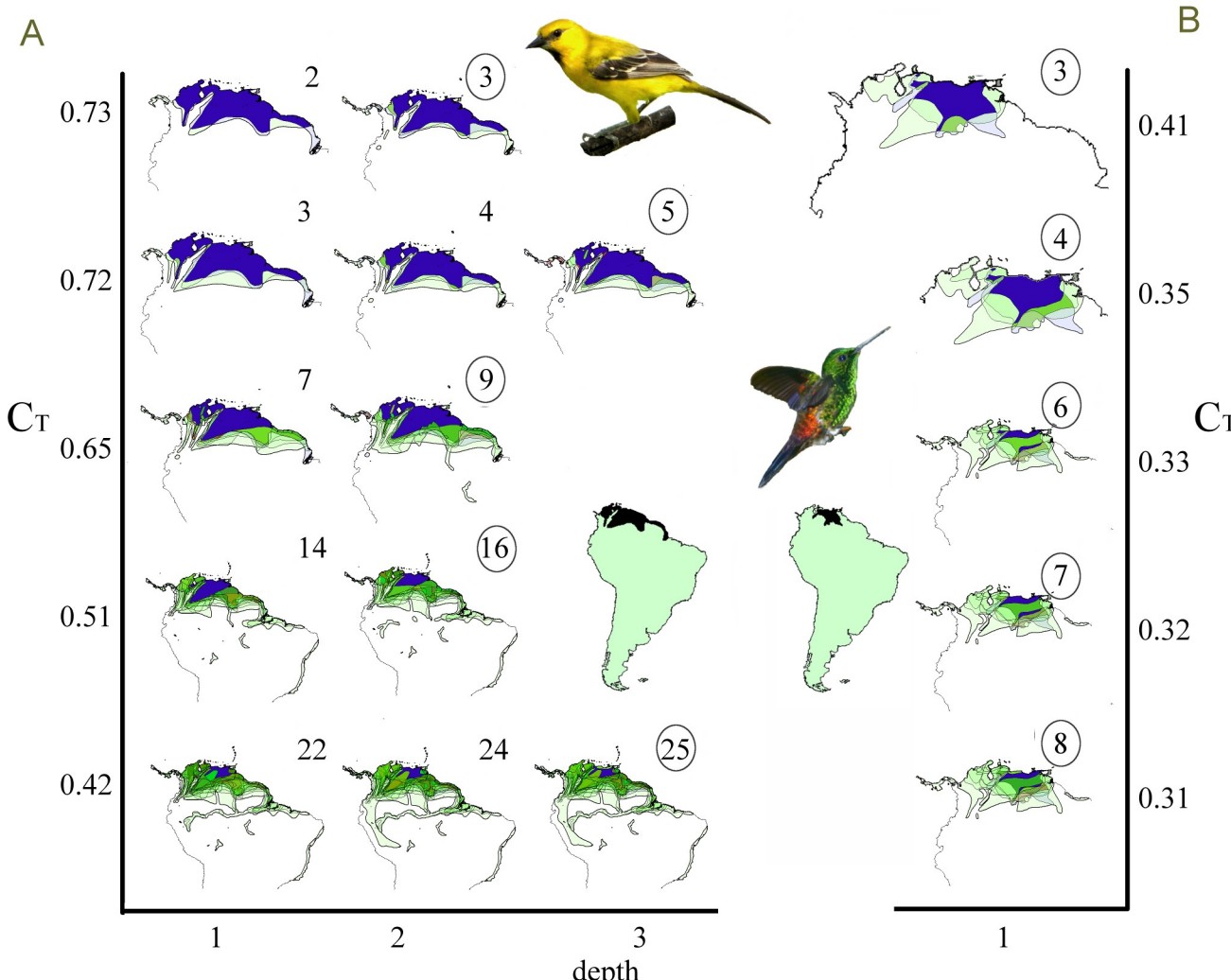

**Fig 4. Congruence diagrams illustrating the whole set of partial chorotypes of *Icterus nigrogularis* and *Amazilia tobaci* bird species.** A didactic representation of spatial congruence relationships depicts congruence threshold ($C_T$, y-axis, free scale) and depth (x-axis). (A) *Icterus nigrogularis* (Yellow Oriole) and (B) *Amazilia tobaci* (Copper-rumped Hummingbird). For each $C_T$ and depth, ranges (light-green) are overlaid; number of species in each group are shown for each round at each depth. Circles depict the closed lists (partial chorotypes) at the end of each round of threshold analysis. The common area is shown in dark-blue. South America insets show reference species ranges as black. Grouped species and synonymous patterns are listed in S3 and S4 Tables, respectively. Geographic scales differ because maps are zoomed out to accommodate successively larger lists. Credit: images of *I. nigrogularis* and *A. tobaci* adapted from Wikimedia Commons under Attribution-Share Alike 2.0 Generic license (creativecommons.org/licenses/by/2.0/deed.en).

pairwise congruences permeate all steps of the analysis. SCAN then assesses the informative biogeographical potential for each species independently (i.e., whether it generates or composes chorotypes). Also, the algorithm does not classify grid-cells based on their similarities with other cells (e.g., cluster, ordination), but recognizes mutually congruent groups (chorotypes) in a network of species linked (and weighted) by spatial congruences. Degree of congruence is an intuitively simple concept with clear biological relevance. Nevertheless, there is no theoretical basis for establishing any specific numerical threshold *a priori*, but the method permits the use of relaxed settings to explore all potential partial chorotypes derived from the web of spatial relationships of each reference species. Chorotypes are groups of species, not specific regions, and the final delimitation of the taxonomic composition and geographic area they

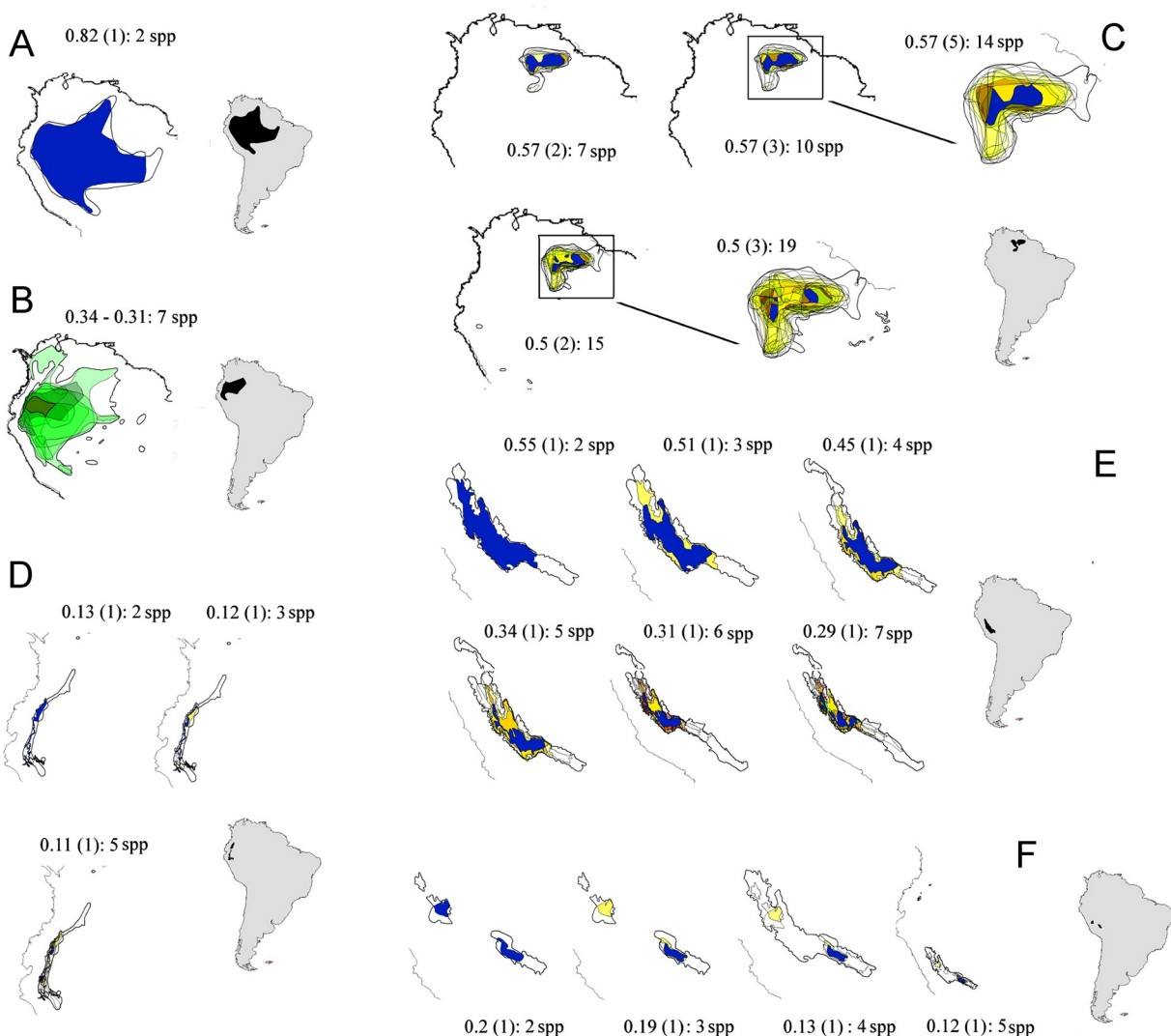

**Fig 5. Partial chorotypes of *Heliodoxa* hummingbirds in lowland Amazonia and pre-Andes.** Brilliant hummingbirds include a large diversity of species with distinctive biogeographical characteristics. Widespread lowland species: (A) *H. aurescens* (Gould's Jewelfront) composes a two species chorotype, and (B) *H. schreibersii* (Black-throated Brilliant) is not biogeographically informative (does not gives rise to closed lists). Highland species show spatially localized patterns. In the east, the Pantepui chorotype (C) *H. xanthogonys* (Velvet-browed Brilliant). In the west, the pre-Andean region: (D) *H. gularis* (Pink-throated Brilliant); (E) *H. branickii* (Rufous-webbed Brilliant); (F) *H. whitelyana* (Black-breasted Brilliant). These summarized views of chorotypes present the range of congruence thresholds, depth (in parentheses), and number of species as descriptive parameters. Grouped species are listed in S3 Table. Ranges of individual species leading to partial chorotypes are shown in yellow; the common area for all species is in blue.

represent can be selected among alternative configurations, based on environmental or geographic criteria, or specific objectives of the study. Finally, SCAN correctly recognizes and describes gradients of range distributions. Despite the key role gradients have had in the development of ecological thinking [48, 49], they have not received the same attention in the field of biogeography [13, 50, 51], which tends to see imperfect congruence as an inconvenient deviation from idealized responses to geographic barriers. Using SCAN, then, should allow increased detection of and attention to distributional gradients.

The main characteristics of the method were demonstrated by the data sets analyzed. These analyses showed that SCAN generates results that may be indicative of possible spatial drivers

of species ranges. The richness in details and alternative results of the simulated gradient reinforce the idea of a high "biogeographical resolution" of SCAN results. The bird dataset showed that idiosyncratic patterns derived from real life species, presenting highly distinctive alternative configurations, can be recovered in detail across a range of congruence thresholds. This apparent contradiction is better seen as evidence for the co-existence of distinct but complementary phenomena driving distributions of spatially related species. Large chorotypes containing numerous and varied partial chorotypes, for instance, may carry information about vicariant causes in small highly congruent partial chorotypes, but also about processes, such as local extinctions, dispersal events, and differential responses controlled by species-specific traits [50, 52], responsible for the formation of spatial gradients, recovered under relaxed thresholds. Fuzzy entropy, which indicates degree of complexity in chorotypes and of range border coincidence among species [17, 53], offers an analogous approach. Fuzzy patterns with low entropy values, potentially driven by historical causes, can be compared to highly-congruent partial chorotypes identified by SCAN. Conversely, SCAN's complex chorotypes combining nuclear areas expanded through gradients identified by indirect relationships would correspond to fuzzy chorotypes with high entropy.

Many of SCAN's properties are derived from the objective focus on species ranges and their spatial congruences; these relationships are crystallized in SCAN's one-layered networks, where species are vertices (or nodes) connected by their respective spatial congruences (edges). The algorithm proposed here to traverse the network is a type of "Breadth-First Search (BFS)" self-stabilizing spanning tree [54], in which all vertices connected to a reference species are analyzed at the first depth level before starting the analyses of indirect connections, at each subsequent depth level. This level-by-level approach characterizes BFS, as opposed to "Depth-First Search", which explores branch connections in full depth before analyzing all neighbors in the same level. In SCAN, edges (connections) are weighted by $C_S$ estimates. The algorithm, however, evaluates only a subset of the whole network (not the full set of connecting overlaps as in other methods) composed of pairs of ranges matching the congruence threshold being evaluated. In practice, each round analyzes only the direct and indirect links which are "activated" at this respective $C_T$. This conditional connection assures that spatial congruences based on whole ranges are considered during algorithm processing, and that all potentially connected species have comparable levels of spatial congruence. When a group closes, the resulting patterns are biogeographical units built over pure species-spatial inter-relationships, analogous to the concepts of 'community' or 'module' in network analysis, that is, sets of nodes that are more densely connected to each other than to other nodes of the network [55]. The biogeographical units recognized through SCAN are analogous to the global chorotypes of Fattorini [4, 15], the fuzzy chorotypes of Olivero et al. [17], and the generic definition of biotic elements of Hausdorf [25]. The partial chorotypes are conditional to congruence thresholds. Partial chorotypes can also be evaluated and compared based on metrics (see Fig 4), such as the number of species, depth, and degree of spatial coherence (e.g., the ratio between common and total areas).

## Patterns, parameters, and biological correlates

Examining the biological meaning of SCAN's parameters calls attention to important trade-offs in pattern recognition. For example, high species richness in a spatial pattern, that is, its inclusion of numerous species, is a desirable feature for any pattern. However, it is usually achieved at the expense of spatial coherence, because more species are usually grouped with lower congruences or greater depths, which in turn lead to smaller common and larger total areas. In biotas dominated by idiosyncratic ranges, small gains in species richness may quickly

disfigure the spatial coherence of a pattern. Conversely, a regional community shaped by effective barriers (e.g., [56]) or by zones of intense environmental filtering [57], should display patterns that add species while maintaining a certain level of spatial coherence.

The concept of depth appears to have no clear biological analog. In practice, however, it allows the identification of gradients through indirect spatial relationships, in a delicate equilibrium. As seen in the simulated gradient, the chain of links must be limited by a discontinuous interval, which closes a group. This paradox (connection vs. disconnection) is the essence of SCAN: regardless of the congruence threshold, if there are closed groups, then there is spatial cohesion among their constituent taxa relative to the pool. Under low congruence thresholds and large maximum depth settings, however, links may potentially lead to distortions in range sizes, which often prevent a group from closing (see Fig 5B).

Finally, the assessment of the biogeographically informative status of a species, from the chorological perspective, may open a path to the investigation of phyletic or ecological correlates to the informative (or not) status of a given species. This information has the potential to improve spatial classification attempts, in the domain of the regionalization approach. Floras and faunas, in particular those of mega-diverse tropical regions, often present a high proportion of idiosyncratic historic and spatial responses (e.g., [50]). Depending on specific purposes and objectives, the selection of biogeographically meaningful species, under a chorological perspective, has the potential to minimize bias caused by redundant or conflicting spatial information [58]. The application of fuzzy characterization (sensu [17]) of amphibian [37] and bird [53] chorotypes already showed how the chorological interpretation of patterns may be used to improve the identification of biogeographical regions and transition zones.

## Concluding remarks

The development of this method was stimulated in part by the desire to minimize assumptions, particularly those related to the biological causes of spatial congruence in ranges and of pattern formation (e.g., vicariance, dispersal, ecological determinism). However, like any model that sets out to describe real-world patterns, some general assumptions apply to this framework. We recognize two. First, species ranges and their direct and indirect relationships are biogeographically meaningful; this is an assumption of all analogous biogeographical analyses. Second, range maps are good representations of such distributions, at least at the spatial scale of interest; this is a potential weak point, in that current range maps in certain regions are notoriously flawed [59]. However, poor distributional data, especially data suffering from chronic omission (e.g., [60]), will also affect all other methods, and improvements in maps and taxonomy can be expected over time.

The richness of patterns detected by SCAN is a reflection of natural phenomena and natural groupings of species. However, the complex webs of spatial interrelationships revealed require more effort to organize and interpret than results of other methods. The description of biogeographical scenarios depends on posterior identification of synonymous and nested chorotypes, for example, such as were encountered in both datasets examined in this study. Rather than a drawback, this post-algorithm analysis can be viewed as an opportunity to better understand these patterns and their variations, species properties, distinct spatial outlines and their environmental correlates, and to make explicit the objectives and assumptions of a particular investigation. For simple chorotypes, the interpretative options are limited (e.g., Fig 4B). For larger ones, the amount of complementary information may constitute a real challenge (e.g., Figs 4A and 5C). SCAN asks the biogeographer to interpret individual chorotypes as a suite of plausible biogeographical scenarios. Highly congruent partial chorotypes highlight potential historical barriers, or sharp ecological limits or interactions. Patterns of relaxed congruence

may reveal taxonomic or environmental correlates of selective permeability to barriers, and potential dispersal routes. Trade-offs and criteria of spatial coherence and comprehensiveness may also be complemented by biological or environmental information to support both spatial classifications and assessments of spatial drivers. The approach proposed here describes possible scenarios based on congruent species distributions–the burden of biogeographical interpretation and attribution of processes to explain the patterns is independent of the analysis and lies with the investigator.

## Supporting information

**S1 Fig. Schematic overview over the spatial congruence framework.** (A) Spatial representation of a hypothetical regional community of ten simulated species ($c1$-$c10$). (B) Congruence similarities between all overlapping species pairs ($C_S \geq 0.01$) are used for all direct and indirect comparisons in the following steps. (C) At relatively high congruence thresholds ($C_T$), as references species, $c1$ and $c6$ each gives rise to only one partial chorotype (closed list) composed of two directly-related species ($C_T = 0.67$ and $0.71$, respectively). (D) Depending on $C_T$, $c4$ may have numerous indirect relationships. At $C_T$ between 0.46 and 0.22, it is only (directly) connected to $c9$. As of 0.21, $c4$ is also directly connected (depth 1) to $c8$ which links (indicated with arrow) to $c6$ (depth 2), which in turn links to $c7$ (depth 3). This partial chorotype (indicated with an asterisk) will be further examined in subsequent examples. At 0.19, an additional link, between $c9$ and $c3$, appears at depth 2. However, with this suite of ranges, there is no longer any area of overlap among all ranges (shown in black when it occurs); the lack of range overlap would cause the analysis to drop this $C_T$ round. (E) All informative reference species and their respective partial chorotypes recovered (colored as in preceding figures). Ranges $c5$ and $c10$ were not included in any patterns, and $c3$ was included in some groups, but did not give rise to partial chorotypes. The pattern [$c4$+$c9$+$c8$+$c6$+$c7$] can be derived from any of its constituent species (dark blue; asterisk). (F) Possible classification schemes show the trade-off between congruence and comprehensiveness. A highly congruent scheme has three independent two-species patterns (blue, green, and red patterns), and $c3$ and $c8$ are out (yellow). A less congruent scheme (asterisk) has a very comprehensive pattern (blue) based on less congruent indirect links, with a common area for all ranges. It encompasses many nested sub-patterns (shown as distinct colors in the previous classification).
(TIF)

**S2 Fig. Histograms of pairwise range overlaps among the whole avifauna, informative taxa, and (partial) chorotypes.** When all range overlaps of the avifauna are considered (first row), most have low spatial congruence similarity values (first column; mean±sd = 0.13±0.18, n = 1.37M overlaps), a high number of overlaps by species (second column; 1252±520, n = 1095 species), and low CS means (third column; 0.12±0.06, n = 1095 species). The same pattern is repeated for all overlaps of informative species (second row; n = 519 species). When only overlaps composing chorotypes are considered (third row) the pattern is inverted. $C_S$ values are high (0.71±0.17; n = 14597 overlaps of 558 species) for a small number of overlaps (26 ±34, n = 558) with higher mean $C_S$ (0.61±0.19, n = 558). When widespread and non-Amazonian chorotypes are filtered out (forth row) the pattern is modified with slightly lower means (0.57±0.2, n = 5088), overlaps (11.1±12.8, n = 457), and $C_S$ (0.57±0.18,n = 457).
(TIF)

**S3 Fig. Exploratory analyses of the relationships between range overlaps, spatial congruence, and chorotypes among lowland birds of Amazonia sensu latissimo [48].** From (B) to (J) two disproportionately large patterns (one gathering 99 transcontinental, and another 92

widespread taxa all across Amazonia) were filtered out. In general, ordinary linear regression analyses (OLS) showed that taxa with larger ranges have higher $C_S$ means (A, $r^2 = 0.66$, $F_{1,1093} = 2099$, p<0.01), even for partial chorotypes (B, $r^2 = 0.5$, $F_{1,388} = 392$, p<0.001). However, patterns derived from larger ranges are not richer (C, $r^2 = 0$). The richness of a chorotype is not related to the mean $C_S$ (D, $r^2 = 0$), and only negligible effects from spam (E, $r^2 = 0.05$, $F_{1,455} = 26$, p<0.001) or maximum threshold values ($C_T$) were detected (F, $r^2 = 0.11$, $F_{1,455} = 56$, p<0.001). For all chorotypes, $C_T$ has a negligible positive effect over the number of species and mean depth (not shown, $r^2 = 0$). However, if reference species are modeled as random effects in linear mixed models, lower $C_T$ allow more species (G, Fixed effect: accumulated number of species Ct, r2 = 0.84, F1,4473 = 2780, p<0.001; Random effect: reference species, Var = 157.9 ±12.1, Res = 15.8, Var ratio = 9.98, 90% explained), at lower mean depths (H, Fixed effect: mean depth, r2 = 0.94, F1,4473 = 472, p<0.001; Random effect: reference species, Var = 1.89 ±0.13, Res = 0.11, Var ratio = 17, 94.5% explained). Depth is positively related with species richness (I; OLS, r2 = 0.53, F1,4493 = 5035, p<0.001). Some data distributions may violate premises of OLS regressions, such as heteroscedasticity: we opted to kept these distributions as they are (i.e., non-transformed) to allow a better visualization of these distributions in their natural shapes. In this context, statistical significance and regression coefficients may be seen only from an exploratory perspective.
(TIF)

**S1 Table. Partial chorotypes in the simulated hypothetical gradient.** Among the 30 'species' analyzed as references, the algorithm recognized 27 as biogeographically informative. Although smaller, the maximum depth setting of 3 (Max-depth) allowed the recognition of highly congruent patterns, as shown by the maximum and minimum congruence thresholds. These partial chorotypes are enough to classify the gradient into 5 distinct non-overlapping zones (Fig 3A), which gain more species and expand at lower congruences (Fig 3B). More relaxed depth settings allow larger groups at intermediate congruences with larger chains of indirect connections. All 23 unique partial chorotypes recovered are nested to one of the patterns grouping all species of the South or North (11 and 12 patterns, respectively; Fig 3D). Partial chorotypes depicted in Fig 3 are referenced in the last column.
(RTF)

**S2 Table. Simulated hypothetical gradient of ranges by maximum depth settings.** All possible partial chorotypes for each reference species at each distinct max-depth configuration (3, 5, 7, and 10). Number of species, spatial congruence similarity, and max. and min. values of threshold and depth for each chorotype are presented.
(RTF)

**S3 Table. Partial chorotypes derived from selected illustrative South American bird species.** The congruence algorithm groups species with direct (depth 1) and indirect (depth > 1) relationships to the reference at each congruence threshold ($C_T$). $C_S$ index shows the lower $C_T$ which allows a direct relationship between the grouped species and the reference. $C_T$ max and min show the range of $C_T$ values in which the group species compose the partial chorotype. For example, the *Icterus nigrogularis* chorotype at $C_T = 0.73$ ($C_T$ max) (follow $C_T$ max and depth values at Fig 4A) has *Tyrannus dominicensis* directly related ($C_S = 0.73$); this species links the reference to *Hydropsalis cayennensis* (depth = 2), which has a direct congruence to the reference of only $C_S = 0.71$ (lower than the current threshold). At $C_T$ max = 0.42 the last species joins the group (*Inezia caudata*). Taxonomy and distribution follow BirdLife and HBW [46].
(RTF)

**S4 Table. Bird's synonym and nested chorotypes.** Species sharing biogeographical properties converge to synonym chorotypes, in which parameters, and spatial and taxonomic composition are mostly alike. Nested chorotypes are taxonomic subsets of a larger pattern. The patterns shown at Figs 4 and 5 (references species in bold) can be generated by other species grouped at their respective chorotypes. Usually the higher $C_S$ mean to all other members gives the name to the chorotype. The only exception is *Heliodoxa xanthogonys*, which here, for convenience, gives its name to the 'Tepuis1' pattern. This is the only presented chorotype with nested subsets (1.1, 1.1.1). For patterns at the Amazonian margins, not all species composing the groups (S1 Table) matched the criteria of inclusion, and were not analyzed as references (e.g, some *Heliodoxa* spp. chorotypes).
(RTF)

**S1 Script. Source code to the congruence framework functions.** This script is meant to be maintained and updated at https://github.com/cassianogatto/congruence_source. To use this version, save the following script (code lines) as a 'source_congruence_1.1.R' document in the main directory of your analyzes, and 'source' it using the following code: "source ("C:/my_directory/. . ./source_congruence_1.1.R')"; to edit the script load it on an editor, such as RStudio https://rstudio.com/products/rstudio/).
(RTF)

**S2 Script. A brief tutorial to apply the congruence framework to the theoretical simulated gradient of ranges of Kreft & Jetz (2013) [23].** The tutorial and the source script are better viewed in a code editor, such as Rstudio (www.rstudio.org). The functions presented at the Tutorial S2.1 are running and integrated, but the whole framework is still in early stages of code development. Some functions are auxiliary tools, such as coherence_to_sp, which uses congruence and depth to plot range relationships in a customized way (indicating 'internal', 'external', and other spatial relations). Many others are intermediary tools called by higher hierarchical functions. New versions will maintain the functionality but the code will be refined, re-organized as a package, and uploaded opportunistically at https://github.com/cassianogatto/congruence_source (current is congruence_source_1.1.R).
(RTF)

## Acknowledgments

This paper is dedicated to Professor Cláudio J Barros de Carvalho. We acknowledge the invaluable comments and suggestions of two anonymous reviewers, and of the academic editor Bruno Bellisario. We are grateful to Camila C Ribas for valuable comments on earlier versions of the manuscript, and all the staff at the Programa de Pós-Gradução em Ecologia at Instituto Nacional de Pesquisas da Amazônia—INPA. Joel Cracraft and Thiago SF Silva kindly criticized many of CG's early unconventional ideas and hosted CG at their laboratories in the American Museum of Natural History—AMNH, NY, and Unesp-Rio Claro, São Paulo, Brazil, respectively. This paper resulted from CG's Ph.D. dissertation in INPA's Graduate Program in Ecology and is contribution 58 in the Amazonian Ornithology Technical Series of the INPA Biological Collections Program. We have no conflicts of interest.

## Author Contributions

**Conceptualization:** Cassiano A. F. R. Gatto.

**Data curation:** Cassiano A. F. R. Gatto.

**Formal analysis:** Cassiano A. F. R. Gatto.

**Investigation:** Cassiano A. F. R. Gatto.

**Methodology:** Cassiano A. F. R. Gatto.

**Software:** Cassiano A. F. R. Gatto.

**Writing – original draft:** Cassiano A. F. R. Gatto, Mario Cohn-Haft.

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
