## [Decision Letter · Decision Letter 0]

11 Feb 2021

PONE-D-21-00528

Spatial Congruence Analysis (SCAN): An objective method for detecting biogeographical patterns based on species’ range congruences

PLOS ONE

Dear Dr. Gatto,

Thank you for submitting your manuscript to PLOS ONE. After careful consideration, we feel that it has merit but does not fully meet PLOS ONE’s publication criteria as it currently stands. Therefore, we invite you to submit a revised version of the manuscript that addresses the points raised during the review process.

As you can see from referees ‘comments, your paper needs a major revision before being considered for publication. 

I really enjoyed reading your paper, as I believe has the potential to provide robust tools to be employed in biogeographic studies, introducing a formal approach (mostly based on automatic procedures) able to disentangle the (possible) mechanisms behind species’ distributional patterns. However, I strongly agree with both referees about the need to clarify the use of specific (and correct) “biogeographic” concepts that, otherwise, might hinder the real potential of the ms. Moreover, I agree with a clear justification on the use of SCAN with respect to available alternatives, which would allow for a more in-depth justification of the proposed method.

We look forward to receiving your revised manuscript.

Kind regards,

Bruno Bellisario, PhD

Academic Editor

PLOS ONE

Journal Requirements:

Reviewers' comments:

Reviewer's Responses to Questions

**Comments to the Author**

1. Is the manuscript technically sound, and do the data support the conclusions?

Reviewer #1: Yes

Reviewer #2: Yes

2. Has the statistical analysis been performed appropriately and rigorously? 

Reviewer #1: Yes

Reviewer #2: Yes

3. Have the authors made all data underlying the findings in their manuscript fully available?

Reviewer #1: Yes

Reviewer #2: Yes

4. Is the manuscript presented in an intelligible fashion and written in standard English?

Reviewer #1: Yes

Reviewer #2: Yes

5. Review Comments to the Author

Reviewer #1: This paper presents an interesting approach to identify chorotypes, i.e. groups of species with similar distributions. So far, chorotype identification has been done mostly on intuitive grounds. So, I welcome the introduction of a formal approach based on a flexible automatic procedure.

A second important merit of the paper (which, however, is not fully recognized by the authors themselves) is that their approach allows the possibility of inferring some explanatory interpretation to pattern congruence. So far, chorotypes have been largely used as merely descriptive tools. With this new approach, we can shed some lights into the possible mechanisms that produced more or less congruent distributional patterns.

However, the paper has a very serious flaw. The authors mix two conceptually completely different problems: (1) the identification of groups of species with similar ranges and (2) the identification of biogeographic regions. These are epistemologically separate research programs. The first one deals with species (with the aim of grouping species on the basis of their distribution), the second one deals with space (with the aim of dividing the space into regions on the basis of the species).

Failure in recognizing this basic distinction has generated much confusion in biogeography.

For a discussion on this difference, the need of avoiding confusion and the problems that such confusion has generated in biogeography see Fattorini (2016)

The method proposed by the authors deals with the first program, not with the second. Thus, I strongly recommend removing all parts of the manuscript where the method is associated or used for the second aim. This only creates confusion, makes the paper unnecessary long and does not allow clear understanding of paper merits.

Thus, I recommend:

- Delete lines 85-96: These lines refer to methods for biogeographic regionalization, to to species groupings

- Delete lines 104-105: This creates confusion, as mix the two concepts of searching for species groups and searching for regions. Delete.

- Delete lines 375-410: All this section presents an application of the method to biogeographic regionalization, while the method deals with species grouping. All this part is misleading.

-Delete lines 415-416, 422-23, 427-30, 477-79, 499-508: All these parts make confusion between the two concepts.

Lines 16-17: delete “and so form the basis for biogeographical concepts such as areas of endemism and ecoregions”

18-19: delete “much less incorporated in bioregionalization methods as an explicit parameter”

30-31: delete “without confounding transition zones with true biogeographical units, a frequent pitfall of other methods”

42: delete bioregionalization and areas of endemism. Use the keywords: aerography and chorotype

For the same reasons I suggest to change the title as follows:

Spatial Congruence Analysis (SCAN): An objective method for detecting species’ range congruences

I strongly recommend using “chorotype” in the abstract.

In the introduction the authors should also clarify that their approach leads to the identification of regional, not global chorotypes, unless data distribution covers the entire species’ ranges (for this distinction see Fattorini (2015).

Another issue is about the use of the expression “biogeographic element”. This expression has been used with different meanings and it usually refers to a concept very different from that of chorotype. Failure in distinguishing chorotypes from elements has originated much misunderstanding (see Fattorini 2017). Thus I recommend NOT using this expression to identify species groupings as in line 118. I think that such groups identified by the algorithm might be called “closed lists”.

Also I don’t think they correspond really to Hausdorf’s “biotic elements” as biotic elements sensu Hausdorf belong to the same area of endemism - see Hausdorf and Hennig 2003, p. 717), whereas this is not implied in the concept of chorotype (see also line 23 and other occurrences).

Minor comments

Language

I suggest adopting a more direct language, avoiding redundancies

51: delete “surely”

53: delete “in nature”

54: delete: “the study of”

55: “biogeography, because it may be possible to infer from it the action of shared” -> “biogeography, making it possible to infer the action of shared”

57: delete: “to this day”

412-13: Unnecessary sentence.

437: delete “and can be used to make biogeographical decisions.”

Clarity

Some points are not fully explained

Line 305: what numerical values correspond to “slightly lower congruence thresholds”?

Other suggestions/corrections

59: place ref # 4 after “chorotypes”

61: precisely -> very

73: collecting stations, characters -> collecting stations uased as characters in a

366: schreibersii (lower case)

367: xanthogonys (lower case)

695: check n=1.37M

714-19: use exponents for 2 in R2; use subscripts for degrees of freedom in F. Explain how the anaklyses were conducted. I assume that those of lines 706-13 and 719 are OLS regressions. But what about those of lines 713-18?

From figure S3 it is apparent that data on panels A and B are heteroscedastic. I suggest log-transforming them prior to analysis.

Figure 3: explain what the numbers 0.89, 0.94, 0.76 etc mean

Fattorini, S. (2015). On the concept of chorotype. J Biogeogr., 42: 2246–2251.

Fattorini, S. (2016). A history of chorological categories. History and Philosophy of the Life Sciences 38 (3).

Fattorini, S. (2017). The Watson-Forbes Biogeographical Controversy Untangled 170 Years Later. J Hist Biol. 2017 Aug;50(3):473-496. doi: 10.1007/s10739-016-9454-7.

Reviewer #2: “Spatial Congruence Analysis (SCAN): An objective method for detecting biogeographical patterns based on species’ range congruences” is a well written manuscript, which exposes in a clear manner a methodological proposal with applications in biogeography. SCAN provides biogeographers with a tool for addressing the analysis of biogeographic patterns without the need of assumptions regarding to the causes of these patterns. The authors also conceive a level of complexity in nature according to which discrete patterns, such as types of distributions shared by groups of species, may coexist within the same area with species distributions that cannot be biogeographically clustered. The authors propose a method that, in the title of the manuscript, is presented as objective.

However, I have some concerns about this manuscript in its present state. The most important one is related to the innovation of the method, which cannot be evaluated in the absence of comparisons with available alternatives. Objectivity, the avoidance of a-priori assumptions, and the awareness of biogeographic complexity have been objectives in previous proposals. However, the reading of this manuscript transmits the idea that these aims are main achievements of the SCAN. In my opinion, this method provides innovation enough to be published; but references to previous works that shared objectives and achievements must be included. Only then the real extent of the innovation will be perceived.

BIOGEOGRAPHIC NOMENCLATURE

I find a bit of confusion in the use of nomenclatures that are employed in this manuscript, some of which have equivalences in the normal use of biogeography:

The authors describe patterns that are named “biogeographic elements” (lines 118, 221…), “biogeographic complexes” (lines 125, 224…) and “synonyms” (lines 224…). These patterns are clearly defined in the text, and the relationships between each other are also well explained.

The authors state that a “biogeographic element” is analogous to Hausdorf’s “biotic element”: that is, “a group of taxa whose ranges are significantly more similar to each other than to those of taxa of other such groups”. Among the different definitions of “biotic element” [Morrone (2014) Systematics and Biodiversity 12:382-392], Hausdorf’s may reflect, indeed, the most usual meaning of it.

As I see it, “biogeographic elements”, “biogeographic complexes” and “synonyms” are steps along the procedure that drives to the final output, that is, to “biotic elements”. These biotic elements could sometimes correspond to synonyms, and other times to biogeographic elements that do not have synonyms. So, the use of “biotic element” for referring to the final output would be helpful for avoiding the [already existing] nomenclatural inflation in biogeography.

GRID APPROACH VS. POLYGON APPROACH

The authors recognize that “indexes and grid-based analyses have numerous computational advantages” (line 74). In contrast, they underlie the fact that this approach makes grids the information-bearing unit (line 75), which makes the pattern detection be subject to scale bias (line 90). These are true statements. However, I find little support for the statement that, using a grid approach, “patterns that unite species based on similarities in their total ranges may actually become harder to detect” (line 76). The authors provide an example in which, supposedly, only a method like SCAN, looking for both direct and indirect congruences between species distributions, could detect meaningful patterns: “species that shared a historical center of dispersal, but dispersed out of that area to different extents might show very different overall ranges around a common area” (lines 78-84). Desirably, indirect congruences should be accepted up to a reasonable limit (see lines 194-199). Otherwise, barely-overlapping species forming part of a same pattern would constitute outrageous paradoxes. Independently of the method employed, the relative contribution of a common centre of dispersal, compared to that of factors leading to dispersal routes, should be in line with the degree of congruence/similarity between the distributions involved in the process. Among the computational advantages of grid approaches, there are ways to calibrate requirements regarding to the similarity between distributions forming part of the same pattern (e.g. according to statistic criteria). Finally, whichever the approach employed, it is always possible to address the search for links, a posteriori, between different but not independent patterns (e.g. hierarchical grouping between biotic elements potentially denoting causal factors acting at different levels). In conclusion, both a grid approach and a polygon approach like SCAN have potential for deep exploration in complex biogeographical patterns.

“NOT MEANINGFUL” SPATIAL RELATIOSHIPS

I strongly agree with the authors’ view of a spatial coexistence between discrete patterns (represented by biotic elements) and species whose distributions do not match the biotic-element concept (and instead show a gradual way of overlapping). However, I find inconsistent the author’s assumption that “species that show no meaningful spatial relationships with others […] unless further taxonomic or distributional updates become available […] may be thought of as biogeographically uninformative” (lines 267-269). When addressing the a-posteriori search for causal processes (line 36), the researcher could explore the historical and ecological bases of a given biotic element; and could also be interested on the drivers of gradual patterns, e.g. searching for consistencies with environmental gradients or with still-active dispersal processes. So, the existence of species that do not match the discrete view of biogeographic patterns might be plenty of historical and/or ecological meaning.

The authors “test the effect of inclusion of uninformative species on methods of spatial classification” (line 269), by comparing bioregionalizations that either included or excluded the “uninformative species”. The result of this test shows that the “well known pattern based on species turnover across major Amazonian rivers” is only found when the uninformative species are excluded from the analysis (line 387). In line with this, Figure 6C shows a regionalization in which biogeographic boundaries are represented by sharp environmental ecotones (i.e. the rainforest limits) or by important rivers (i.e. Amazon, Negro, Madeira, Guaporé). This exclusion may have led to the detection of a crisp regionalization that could suggest the presence of barriers to dispersal, which is of high biogeographic interest. However, it should be considered here whether “uninformative species” could provide the bases for gradients with identical biogeographic interest. The authors must recognize the enormous interest of transitional components [see, for example, Williams (1996) Proceedings of the Royal Society of London B 263: 579–588; Morrone (2005) Revista Mexicana de Biodiversidad 76: 207-252]. They wrote in the manuscript: “despite the key role gradients have had in the development of ecological thinking, they have not received the same attention in the field of biogeography, which tends to see imperfect congruence as an inconvenient deviation from idealized responses to geographic barriers” (line 431). I agree, but believe that far from “inconvenient deviations”, the potential existence of gradients and transitions should be considered in biogeographic regionalization. For this aim, there are already methodological approaches that can be employed [e.g. Olivero et al. (2013) Systematic Biology 62: 1-21], in which the species here considered uninformative surely provide valuable information.

NOVELTY OF THE METHOD PROPOSED

In the abstract (line 39), it is said that the SCAN “approach eliminates or reduces limitations of other methods and permits pattern description without hidden assumptions about processes, and so should make a valuable contribution to the biogeographer’s toolbox”. The value of this contribution should be regarded to the methods already available in that toolbox, and here the authors should make a clarification effort.

Congruence vs. similarity:

Mathematically, the “spatial congruence index” (line 135) is the equivalent, in the polygon approach, to Jaccard’s index in the grid approach. Congruence and similarity are, so, synonymous concepts in the search for biotic elements. For decades, similarity has been used in the analysis of biotic elements, under the consideration of its deep biogeographical meaning [e.g. Baroni-Urbani & Collingwood (1977) Acta Zoologica Fennica 152: 1-34]. So, I think that the sentence “degree of congruence (congruence threshold) is an intuitively simple concept LIKELY to have biological relevance” (line 421) should be reconsidered.

Objectivity:

The SCAN procedure is based on the definition of different thresholds (see lines 194 to 199), which values are “preliminarily defined after pilot tests” (line 318). In fact, the authors recognize that “there is no theoretical basis for establishing any specific numerical threshold” (line 423). Although it is immediately said that “the method permits exploring alternatives”, I find here a strong drawback in terms of objectivity. Although it is said in the introduction that the “recognition of congruent distributions has traditionally been a relatively subjective process” (line 60), I see that the SCAN is not free from this fault. Grid approaches, instead, able to deal with theoretical concepts such as critical values for the significance of a similarity index [e.g. Baroni-Urbani y Buser (1976) Systematic Zoology 25: 251-259], have provided methods for the objective detection of biotic elements and chorotypes, while preserving the possibility of gradual patterns overlapping with those chorotypes [e.g. Real et al. (1997) African Journal of Ecology 35: 312-325; Real et al. (2008) Global Ecology and Biogeography 17: 735-746].

Overlap, nestedness and relationships between different patterns:

That “partial spatial overlaps may also occur between independent elements” (line 230) is not new in biogeography. The difference between the analysis of biotic elements and the biogeographic regionalization is that, as the authors explain (line 427) the former “allows the identification of patterns overlapped in space but with distinct species compositions”. This is neither new [see, for example, Birks (1976) New Phytology 77: 257-287; or Baroni-Urbani et al. (1978) Memorie della Società Entomologica Italiana 56: 35-92]. In fact, what is qualified as “the most important take-home message of this paper” by the authors (line 502), that is, “the generic use of bioregionalization methods for spatial classification, recognition of biogeographical patterns, and assessment of their historical and ecological drivers”, is neither a novelty. The combined analysis of regionalization and biotic elements whose drivers are explored a posteriori is a quite explored field [e.g. Birks (1976) New Phytology 77: 257-287; Myklestad & Birks (1993) Journal of Biogeography 20: 1-32; Olivero et al. (2013) Systematic Biology 62: 1-21].

Finally, the SCAN provides other capacities that are of very high interest in biogeography, and this is a remarkable reason for going on with this methodological proposal; but, again, these are not new. One of them is the perception of a link between pattern complexity and causal factor: “highly variable complexes, for instance, congregating from small highly congruent elements to large spatial gradients recovered under relaxed thresholds, may carry information about both vicariant causes, and processes responsible for pattern deconstruction, such as local extinctions, dispersal events, and differential responses controlled by species-specific traits” (lines 442 to 446). Olivero et al. [(2011) Systematic Biology 60: 645-660] and Ferro et al. [(2017) Journal of Biogeography 44: 2145-2160] found, in the high “fuzzy entropy” (i.e. the degree of fuzzyness) of some chorotypes, signs of biogeographic complexity indicating a possible combination of dispersal patterns driven by idiosyncratic responses to ecological factors; whereas a low entropy might indicate a stronger role of history in the chorotype configuration. Another one is in the connection vs. disconnection paradox, “the essence of SCAN: regardless of the congruence threshold, if there are closed groups, then there is spatial cohesion among their constituent taxa relative to the pool” (line 483). This resembles the fact that similarity (as congruence) is a fuzzy concept, and so a similarity value higher than 0 already involves a certain degree of similarity. The consequence of this is that any little overlap between two chorotypes already implies that the species forming the biotic element of one are, to a certain degree, members in the other. This fuzziness in the chorotypes limits may be analysed and can drive to interesting hypotheses on their causes and mutual relations [see Olivero et al. (2011) Systematic Biology 60: 645-660]. However, even in the complexity of biogeographic patterns, the basic units in which the fuzziness is found (i.e. biotic elements and chorotypes) should be delimited according to objectively defined thresholds.

6. PLOS authors have the option to publish the peer review history of their article (what does this mean?). If published, this will include your full peer review and any attached files.

Reviewer #1: No

Reviewer #2: No

---

## [Author Response · Author response to Decision Letter 0]

12 Apr 2021

Manaus, 11 April 2021

Dear editor and reviewers:

We were very pleased with the high quality and insightful content of comments and suggestions. We tried to follow accurately the suggestions and to conciliate the views of each one of the reviewers, and with ours, as well. We acknowledge that the original manuscript had some flaws that compromised understanding of the core ideas presented. We agreed with, and readily accepted most suggestions, with few exceptions which are justified in the commented version of the text.

Following reviewer #1’s suggestions, we made a clear differentiation between chorological and the regionalization approaches to the recognition of patterns based on congruent species ranges. This differentiation was incorporated and used as a standard for the improvement of many aspects along the whole text. In addition, reviewer #1 recommended the exclusion of all parts of the manuscript referring to the regionalization approach, which we did almost completely. However, we maintained the reference to the concept of ‘biogeographically informative species’, that is, those species whose ranges were included in forming chorotypes. In a brief mention in the Discussion, we discuss potential benefits of the use of criteria derived from the chorological approach, such as informative potential, for regionalization procedures, supported by some early examples found in the literature (see Discussion).

The second reviewer brought important aspects regarding the proper contextualization of this proposal in the scientific literature of the field, both in methodological and conceptual aspects. We tried to make more explicit citations to some important ideas in pattern recognition coming from both chorological and regionalization methods and, at the same time, we tried to keep the introduction as light as possible, without making the text denser. We followed suggestions regarding clarification of specific SCAN properties throughout the manuscript, and added some comparisons between SCAN’s characteristics and analogous concepts found in other methods, particularly those concerned with the recognition of chorotypes, such as the fuzzy approach to chorotypes.

Both reviewers converged in their concerns about the terminology used for the biogeographic “entities” described here. Following reviewer #1’s observations, we completely abandoned the use of “biogeographic element” to avoid ambiguities (e.g., chorological vs. homology-based historical interpretations) and adopted the term “chorotype” throughout as suggested. To distinguish what we had called “elements” versus “complexes”, we refer to the former as “partial chorotypes” and the latter simply as chorotypes. The justification is based on the fact that the species composition of these closed lists, conditional to a particular congruence threshold, is a subset of the potential whole set of species which can be grouped at the lowest threshold leading to larger closed groups. The largest partial chorotype is the richest pattern and groups all possible species available in the community analyzed that can be grouped in closed lists. Thus, the set of all partial chorotypes is the chorotype itself. However, we are still open to discuss any other possibility, and suggestions are still welcome.

Finally, we modified the text considerably. The first author’s text editor and reference manager was not able to deal with the heavy archive, with the register of all modifications. To proper show the improvements in the text, we used a commented manuscript, tracking the major modifications paragraph by paragraph, and pointing to detailed aspects in precise comments.

We hope you find the modifications satisfactory enough to validate the publication of this new method. While based on simple concepts, we believe that it can bring some new insights to the search for patterns of shared (congruent) ranges and their respective historical and ecological drivers.

Thank you again for your valuable insights,

Kindly,

Cassiano Gatto & Mario Cohn-Haft

Responses to reviewers’ comments are presented in gray boxes, like this one.

Reviewer #1

This paper presents an interesting approach to identify chorotypes, i.e. groups of species with similar distributions. So far, chorotype identification has been done mostly on intuitive grounds. So, I welcome the introduction of a formal approach based on a flexible automatic procedure.

A second important merit of the paper (which, however, is not fully recognized by the authors themselves) is that their approach allows the possibility of inferring some explanatory interpretation to pattern congruence. So far, chorotypes have been largely used as merely descriptive tools. With this new approach, we can shed some lights into the possible mechanisms that produced more or less congruent distributional patterns.

However, the paper has a very serious flaw. The authors mix two conceptually completely different problems: (1) the identification of groups of species with similar ranges and (2) the identification of biogeographic regions. These are epistemologically separate research programs. The first one deals with species (with the aim of grouping species on the basis of their distribution), the second one deals with space (with the aim of dividing the space into regions on the basis of the species).

Failure in recognizing this basic distinction has generated much confusion in biogeography.

For a discussion on this difference, the need of avoiding confusion and the problems that such confusion has generated in biogeography see Fattorini (2016)

Response: We agree and acknowledge this flaw present in the original manuscript. To rectify this, we introduced a new paragraph (#3) making this distinction clear at the beginning of the manuscript. All following text was adapted to incorporate this differentiation where needed.

The method proposed by the authors deals with the first program, not with the second. Thus, I strongly recommend removing all parts of the manuscript where the method is associated or used for the second aim. This only creates confusion, makes the paper unnecessary long and does not allow clear understanding of paper merits.

Response: We followed the recommendation. Our paper is totally concerned about the chorological program. However, we think that there are common grounds in which both programs can talk to each other. Concepts or techniques conceived for particular objectives may be potentially useful in other contexts. Regionalization attempts, for example, may benefit from refinements in the criteria for the inclusion of species in their analyses, such as the exclusion of species not composing any chorotype. These potential paths for mutual benefits should be explored. Olivero et al. (2013) and Ferro (2017; see references in the manuscript), for example, used a chorological classification and criteria to guide regionalization procedures. Our belief in the potential value of use of regionalization methods for chorological objectives should no longer confuse the objectives of this paper and were greatly reduced and clarified as described for specific cases below.

Thus, I recommend:

- Delete lines 85-96: These lines refer to methods for biogeographic regionalization, to to species groupings 

Response: We did not exclude the full paragraph. Instead, based on the rationale presented above, about a potential for the interchange in specific techniques, we followed a suggestion of the reviewer #2 and added more information on current promising ideas concerned with the recognition of patterns of shared ranges (with potential use in the chorological approach). The endemicity analysis, although described as a method to find areas of endemism, in fact is a tool for the detection of chorotypes, as explained in the text. The second paragraph was fully deleted, following the reviewer’s suggestion.

- Delete lines 104-105: This creates confusion, as mix the two concepts of searching for species groups and searching for regions. Delete.

Deleted

- Delete lines 375-410: All this section presents an application of the method to biogeographic regionalization, while the method deals with species grouping. All this part is misleading.

Deleted the whole session “Non-informative species and spatial classification” and Fig6, as well

-Delete lines 415-416, 422-23, 427-30, 477-79, 499-508: All these parts make confusion between the two concepts.

Deleted

Lines 16-17: delete “and so form the basis for biogeographical concepts such as areas of endemism and ecoregions”

Deleted

18-19: delete “much less incorporated in bioregionalization methods as an explicit parameter”

Rephrased to generalize the sentence to refer to both chorological and regionalization views: “Nevertheless, the degree of spatial range congruence characterizing these spatial relationships is rarely used as an explicit parameter.”

30-31: delete “without confounding transition zones with true biogeographical units, a frequent pitfall of other methods”

Deleted

42: delete bioregionalization and areas of endemism. Use the keywords: aerography and chorotype

Deleted and keywords incorporated

For the same reasons I suggest to change the title as follows:

Spatial Congruence Analysis (SCAN): An objective method for detecting species’ range congruences

Response: the title was modified but we opted to emphasize “congruence” as a criterion to be used in the delimitation of generic ‘biogeographical patterns’.

I strongly recommend using “chorotype” in the abstract.

Done

Response: Most, if not all, of the above recommendations were followed. Exceptional and correlated cases were explained in comment boxes in the corresponding text (see the commented version).

In the introduction the authors should also clarify that their approach leads to the identification of regional, not global chorotypes, unless data distribution covers the entire species’ ranges (for this distinction see Fattorini (2015).

Response: We clarified the distinction between the two approaches in paragraph #3. However, our methods used in this paper actually used global ranges and generated global chorotypes (although SCAN can be used for the identification of regional chorotypes). That the reviewer could misunderstand this clearly was our fault, so we revised the description in the Methods (lines 269) and Discussion (419) to make this clear.

Another issue is about the use of the expression “biogeographic element”. This expression has been used with different meanings and it usually refers to a concept very different from that of chorotype. Failure in distinguishing chorotypes from elements has originated much misunderstanding (see Fattorini 2017). Thus I recommend NOT using this expression to identify species groupings as in line 118. I think that such groups identified by the algorithm might be called “closed lists”.

Also I don’t think they correspond really to Hausdorf’s “biotic elements” as biotic elements sensu Hausdorf belong to the same area of endemism - see Hausdorf and Hennig 2003, p. 717), whereas this is not implied in the concept of chorotype (see also line 23 and other occurrences).

Response: We followed this recommendation and aborted any use of ‘element’ in the chorological approach described here. Closed lists were also named ‘partial chorotypes’, as they carry relevant biogeographical information, but conditioned to a particular threshold. The “complex” (term not used anymore) congregating all partial chorotypes is then referred to throughout simply as the ‘chorotype’. We hope this terminology introduces a kind of hierarchical relationship implicating that the threshold-restricted group is a subset of the whole chorotype. However, we are still open to suggestions of recommendations about this issue.

Minor comments

Language

I suggest adopting a more direct language, avoiding redundancies

51: delete “surely”

53: delete “in nature”

54: delete: “the study of”

55: “biogeography, because it may be possible to infer from it the action of shared” -> “biogeography, making it possible to infer the action of shared”

57: delete: “to this day”

412-13: Unnecessary sentence.

437: delete “and can be used to make biogeographical decisions.”

All recommendations were followed and other similar cases were edited where detected

Clarity

Some points are not fully explained

Line 305: what numerical values correspond to “slightly lower congruence thresholds”?

Other suggestions/corrections

59: place ref # 4 after “chorotypes”

61: precisely -> very

73: collecting stations, characters -> collecting stations uased as characters in a

366: schreibersii (lower case)

367: xanthogonys (lower case)

All recommendations were followed and other similar cases were edited where detected 

714-19: use exponents for 2 in R2; use subscripts for degrees of freedom in F. Explain how the anaklyses were conducted. I assume that those of lines 706-13 and 719 are OLS regressions. But what about those of lines 713-18?

Response: Exponents & subscripts fixed. Yes, those are OLS (now indicated at the legend). The other are GLMM in which each chorotype is a random factor, allowing the capturing of intra-pattern variation, which is opposed to the general trend. 

From figure S3 it is apparent that data on panels A and B are heteroscedastic. I suggest log-transforming them prior to analysis.

Figure 3: explain what the numbers 0.89, 0.94, 0.76 etc mean

Fattorini, S. (2015). On the concept of chorotype. J Biogeogr., 42: 2246–2251.

Fattorini, S. (2016). A history of chorological categories. History and Philosophy of the Life Sciences 38 (3).

Fattorini, S. (2017). The Watson-Forbes Biogeographical Controversy Untangled 170 Years Later. J Hist Biol. 2017 Aug;50(3):473-496. doi: 10.1007/s10739-016-9454-7.

Response: All suggestions were accepted and the text was corrected following reviewer’’s #1 suggestions and recommendations. Fattorini (2016) was incorporated as a reference text regarding the differentiation between chorological and regionalization approaches throughout the text. Any exceptional case was explained in comments in the commented version of the manuscript.

Reviewer #2:

In general, reviewer #2 made challenging and insightful comments rather than specific textual suggestions. Thus, below we respond to these comments one by one at some length. However, in the text of the manuscript itself, many of these issues did not seem to warrant long explanations that would lengthen the paper and perhaps even cloud issues further. Any number of these topics either became moot once certain texts were removed in response to comments from rev #1 (with whom there was considerable concordance) or could be clarified by relatively simple rewording of the text as explained below or in the responses to rev #1 (above). With this in mind then, we have engaged rev #2 in discussion below, but not always incorporated these arguments in the text. We hope that he or she is satisfied both by the discussions below and by the somewhat simplified text in this revision. In any case, we are eager for feedback from this reviewer regarding the adequacy of this new version.

“Spatial Congruence Analysis (SCAN): An objective method for detecting biogeographical patterns based on species’ range congruences” is a well written manuscript, which exposes in a clear manner a methodological proposal with applications in biogeography. SCAN provides biogeographers with a tool for addressing the analysis of biogeographic patterns without the need of assumptions regarding to the causes of these patterns. The authors also conceive a level of complexity in nature according to which discrete patterns, such as types of distributions shared by groups of species, may coexist within the same area with species distributions that cannot be biogeographically clustered. The authors propose a method that, in the title of the manuscript, is presented as objective.

However, I have some concerns about this manuscript in its present state. 

The most important one is related to the innovation of the method, which cannot be evaluated in the absence of comparisons with available alternatives. Objectivity, the avoidance of a-priori assumptions, and the awareness of biogeographic complexity have been objectives in previous proposals. However, the reading of this manuscript transmits the idea that these aims are main achievements of the SCAN. In my opinion, this method provides innovation enough to be published; but references to previous works that shared objectives and achievements must be included. Only then the real extent of the innovation will be perceived.

Response: As PLOS One is a journal with a wide audience, we opted for a more concise and lighter text in the Introduction, with less dense, discipline-specific content. However, we agree that SCAN must be contrasted with previous methods with different properties and similar objectives. In the Introduction we added citations clarifying some above-mentioned aspects and improved sentences in the Introduction and Discussion. We hope that these brief mentions are enough to add more about the background context over which SCAN was developed, while at the same time avoiding confusion among distinct approaches as strongly emphasized by reviewer #1.

BIOGEOGRAPHIC NOMENCLATURE

I find a bit of confusion in the use of nomenclatures that are employed in this manuscript, some of which have equivalences in the normal use of biogeography:

The authors describe patterns that are named “biogeographic elements” (lines 118, 221…), “biogeographic complexes” (lines 125, 224…) and “synonyms” (lines 224…). These patterns are clearly defined in the text, and the relationships between each other are also well explained.

The authors state that a “biogeographic element” is analogous to Hausdorf’s “biotic element”: that is, “a group of taxa whose ranges are significantly more similar to each other than to those of taxa of other such groups”. Among the different definitions of “biotic element” [Morrone (2014) Systematics and Biodiversity 12:382-392], Hausdorf’s may reflect, indeed, the most usual meaning of it.

As I see it, “biogeographic elements”, “biogeographic complexes” and “synonyms” are steps along the procedure that drives to the final output, that is, to “biotic elements”. These biotic elements could sometimes correspond to synonyms, and other times to biogeographic elements that do not have synonyms.

Response: We agree with rev #2 and rev #1 about nomenclatural changes. We liked the point about the steps towards a final output and, definitely, couldn’t agree more. The problem we are facing is how to couple this hierarchical view with terminology. Each closed group generated at each stage has a biogeographical meaning, although each in the context of a specific threshold level of congruence. Inspired by the above comments, we named the closed lists as ‘partial chorotypes’, reflecting their intermediate condition towards the whole pattern (grouping all possible species meeting the ‘closed list’ criterion), the chorotype. A chorotype, thus, refers to the whole complex of partial chorotypes. As explained above in response to rev #1, we adopted this terminology but, at the same time, we are still open to suggestions. 

So, the use of “biotic element” for referring to the final output would be helpful for avoiding the [already existing] nomenclatural inflation in biogeography.

Response: This conflicts with reviewer #1’s comment, so we adopted the term ‘chorotype’, an analogous term that doesn’t require ‘common history’ (as referred by rev #1).

GRID APPROACH VS. POLYGON APPROACH

The authors recognize that “indexes and grid-based analyses have numerous computational advantages” (line 74). In contrast, they underlie the fact that this approach makes grids the information-bearing unit (line 75), which makes the pattern detection be subject to scale bias (line 90). These are true statements. However, I find little support for the statement that, using a grid approach, “patterns that unite species based on similarities in their total ranges may actually become harder to detect” (line 76). 

Response: We agree and the sentence was deleted. Some minor clarification was added to the paragraph – see comments in the commented version of the new manuscript.

The authors provide an example in which, supposedly, only a method like SCAN, looking for both direct and indirect congruences between species distributions, could detect meaningful patterns: “species that shared a historical center of dispersal, but dispersed out of that area to different extents might show very different overall ranges around a common area” (lines 78-84). Desirably, indirect congruences should be accepted up to a reasonable limit (see lines 194-199). Otherwise, barely-overlapping species forming part of a same pattern would constitute outrageous paradoxes. 

Response: Not necessarily. To constitute a pattern (chorotype) under our criterion, species must form ‘closed groups’. Overly extended chains of indirect congruences may be seem to represent an unnatural biogeographic structure. We agree. However, it is always good to have in mind that, to identify those closed groups, the congruence requirements are kept stable and comparable across the chain of indirectly linked species (the congruence to the reference can be very low, but the indirect links are at the same threshold). Furthermore, the relaxation of the ‘overlap criteria’, which stops the algorithm in case any new species is not at least partially overlapped to every other, may allow, at least in theory, extended gradients of ranges even at low overall congruence thresholds. Their mutual congruence, even if at very low levels, are enough to contrast this group in relation to even lower congruences characterizing the neighboring species in the community. Thus, we believe identifying groups and being explicit about the congruence level at which they appear is preferable to prejudging results as outrageous before revealing them and discovering their degree of “connectedness”.

Naturally, we too were worried that low congruence levels or high depth (in indirect congruences) might lead to runaway chorotypes involving virtually all species involved in the study. However, in practice that proved not to be the case, which we believe to be an interesting and important aspect of the method: finding chorotypes at different and explicit levels of congruence. In general, patterns identified at very low congruence requirements were not common but, in the context of scarce neighboring ranges at similar congruence requirements, were indeed possible. See figure 1C for examples of patterns grouping species not directly congruent (not overlapped), but still configuring clear patterns through indirect relationships in gradients. These long gradients are real challenges to most cluster techniques, although ordination methods based on turnover indexes are probably suited for their recognition. Thus, detected them is a strength of SCAN.

Independently of the method employed, the relative contribution of a common centre of dispersal, compared to that of factors leading to dispersal routes, should be in line with the degree of congruence/similarity between the distributions involved in the process.

Response: Couldn’t agree more. We think that this is the essence of SCAN.

Among the computational advantages of grid approaches, there are ways to calibrate requirements regarding to the similarity between distributions forming part of the same pattern (e.g. according to statistic criteria). 

Response: We agree. But, among other limitations, grid-systems in most current standard protocols are not fully suited to deal with indirect relationships (but see network approaches e.g. Vilhena & Antonelli 2015). Another aspect is the independent verification of species spatial connections, one by one, as proposed by SCAN. This facilitates the recognition of highly overlapped independent patterns and allows the comparison of each species with all others. Because not based on grid-cells, SCAN does not use spatial overlap as a criterion for hierarchical nesting, only compositional. If one pattern is spatially merged in another one, they can still be completely independent, with no species in common.

We’ve been thinking about possible ways to apply statistics to SCAN. However, it is still an open subject that we thought premature to include in this initial paper. Our general impression is that, because the overlaps selected in closed groups are highly congruent, hardly compared to ordinary congruence levels of the average overlap, SCAN is statistically ‘redundant’ in a broad perspective. In addition, the indirect connections still pose a challenge to statistical approaches. New ideas or future partnerships to explore this aspect are welcomed.

Finally, whichever the approach employed, it is always possible to address the search for links, a posteriori, between different but not independent patterns (e.g. hierarchical grouping between biotic elements potentially denoting causal factors acting at different levels). In conclusion, both a grid approach and a polygon approach like SCAN have potential for deep exploration in complex biogeographical patterns.

Response: Although we recognize the potential to do such explorations a posteriori using grids/ cluster-analogous techniques, we think that some intrinsic limitations may still hide some spatial relationships. More precisely, many limitations are not associated with the use of grids themselves (as is the case for scale and range distortions), but with some of the standard cluster (and analogous) protocols. The Endemicity Analysis (Szumik et al. 2002) and Infomap Bioregions (Vilhena & Antonelli 2015) are two examples of grid-methods with unique capabilities. The first method recognizes chorotypes (although it is used to search for ‘areas of endemism’ – their definition of AE resembles the definition of a chorotype, but they assume common history – this is an endless discussion). It is a precise method to evaluate ‘independent’ spatial relationships, as SCAN but, as the algorithm selects areas, the synonymization is between areas. But it cannot detect gradients of second order (indirect) relationships. Infomap Bioregions is capable of recognizing such indirect relationships through random walks in networks, although it is not clear if the use of these indirect relationships is implemented in the current version (see www.mapequation.org/Bioregions). Cluster, as a special case of network analysis, uses all spatial information at the same time in every analysis. Even optimal adjustments in parameters can hardly balance the fine resolution required to identify highly overlapped, but distinct units, among all cluster groups in a large community and, at the same time, identify groups of species that are related in very relaxed congruence relationships. Even knowing that fuzzy logic can detect gradients between different units, some natural chorotypes (following the criteria used here) would require more relaxed, some more restricted settings to be detected. Moreover, Infomap Bioregions uses parameters that are strictly related to the regionalization analysis itself (number of trials, cluster cost), hardly being directly interpreted in biological terms. SCAN combines both capabilities, independent assessment and indirect detection, using spatial congruence as the common language – to our knowledge, this is a novelty.

“NOT MEANINGFUL” SPATIAL RELATIOSHIPS

I strongly agree with the authors’ view of a spatial coexistence between discrete patterns (represented by biotic elements) and species whose distributions do not match the biotic-element concept (and instead show a gradual way of overlapping).

Response: In our criterion, any species composing biotic elements (now called “partial chorotypes” or “closed lists”), be it as reference species, or directly or indirectly linked to any reference species in a closed list, is considered informative. Among these species, some may configure gradients of ranges, particularly those indirectly connected at deeper levels. Thus, gradients are contemplated by the criterion of closed lists (partial chorotypes – see hypothetical possibilities in Fig 1C). “Gradual way of overlapping” may or may not represent a chorotype, and SCAN will recognize if the species configuring the gradient share this characteristic among themselves but not with other species (other community members not enlisted to this specific pattern), at particular threshold levels. Further examination of chorotypes may indicate a ‘meta’ gradient in patterns recovered as independent chorotypes (we found evidence for this ‘meta’ arrangement of chorotypes in our second paper describing Amazonian patterns in detail, which is still in preparation).

However, I find inconsistent the author’s assumption that “species that show no meaningful spatial relationships with others […] unless further taxonomic or distributional updates become available […] may be thought of as biogeographically uninformative” (lines 267-269). When addressing the a-posteriori search for causal processes (line 36), the researcher could explore the historical and ecological bases of a given biotic element; and could also be interested on the drivers of gradual patterns, e.g. searching for consistencies with environmental gradients or with still-active dispersal processes. So, the existence of species that do not match the discrete view of biogeographic patterns might be plenty of historical and/or ecological meaning.

Response: We agree in part with this comment but, at the same time, we think that rev #2 misunderstood our concept of ‘non-informative’ species. “Uninformative” are those that do not form any type of closed groups, be it a localized spatial relation in a ‘centre of dispersal’ (usually direct or highly indirect congruences) or species forming gradients of indirect relationships. Thus, they are just those species that might lead to infinitely large and “paradoxical” groupings that the reviewer feared might be revealed. Our non-informative species could be compared to those species not grouped in any chorotype at significant level in the fuzzy analysis of Olivero et al. (2011). So, in fact, if ‘gradient’ patterns are detected, which they often are by SCAN, the species that make up these chorotypes are, indeed, considered informative! As opposed to most methods, our framework identifies and highlights these gradients. 

The authors “test the effect of inclusion of uninformative species on methods of spatial classification” (line 269), by comparing bioregionalizations that either included or excluded the “uninformative species”. The result of this test shows that the “well known pattern based on species turnover across major Amazonian rivers” is only found when the uninformative species are excluded from the analysis (line 387). In line with this, Figure 6C shows a regionalization in which biogeographic boundaries are represented by sharp environmental ecotones (i.e. the rainforest limits) or by important rivers (i.e. Amazon, Negro, Madeira, Guaporé). This exclusion may have led to the detection of a crisp regionalization that could suggest the presence of barriers to dispersal, which is of high biogeographic interest.

Response: Our original intention in this experiment was not to state that the well-known pattern can only be recovered through the exclusion of non-informative species. Instead, we only presented an illustrative example showing that the recognition of ‘informative species’ may lead to clearer and stable results, avoiding the effect of hundreds of idiosyncratic species, in this specific case using the Bioregions framework. The main finding is this “stability” of classifications under distinct cluster and grid-scale settings, which deeply modified the results with the larger species sets. Depending on specific objectives of each study, this filtering of species may be justified. In others, may be not. This general pattern in Amazonia, for example, can be recovered with other methods (e.g., turnover-based ordination; Oliveira et al. 2017 cited in the manuscript). Moreover, among the upland chorotype-associated species there are many species composing gradients. Our intention was to demonstrate how much the choice of species in bioregionalization can modify the results, and that the use of additional criteria (composing chorotypes is an example of criterion) may clarify results that could be not easily interpreted otherwise. We agree that, in general, patterns of terra-firme Amazonian species are delimited, in more or less degree, by major rivers, and this enhances the chances of getting ‘crisp’ borders associated with these rivers. But note that the Amazon river, from central to western portion, was not recognized as a barrier in Fig 6C. In the same way as there are patterns segregated by the river, there are many patterns occupying both margins of the river at this region.

In any case, this entire case study was excluded from the paper. We followed the suggestion of rev #1 about this experiment bringing confusion to the chorotypical x regionalizational approaches and believe we can include it to greater effect in upcoming papers dealing directly with specific applications. The illustration of possible connections and mutual benefits that can be achieved when these distinct programs talk to each other, much like what Olivero et al. (2003) and Ferro et al. (2017) did in their regionalization papers is then briefly cited at the end of the Discussion.

However, it should be considered here whether “uninformative species” could provide the bases for gradients with identical biogeographic interest. The authors must recognize the enormous interest of transitional components [see, for example, Williams (1996) Proceedings of the Royal Society of London B 263: 579–588; Morrone (2005) Revista Mexicana de Biodiversidad 76: 207-252]. They wrote in the manuscript: “despite the key role gradients have had in the development of ecological thinking, they have not received the same attention in the field of biogeography, which tends to see imperfect congruence as an inconvenient deviation from idealized responses to geographic barriers” (line 431). I agree, but believe that far from “inconvenient deviations”, the potential existence of gradients and transitions should be considered in biogeographic regionalization. For this aim, there are already methodological approaches that can be employed [e.g. Olivero et al. (2013) Systematic Biology 62: 1-21], in which the species here considered uninformative surely provide valuable information.

Response: We definitely agree with rev #2. The original intention of the manuscript was to express the exact same view, regarding the importance of the recognition of transitions. In this respect, the use of only informative species may even highlight transition zones, as the identification of species more or less associated to centers of dispersal may avoid some bias caused by species that are, in fact, completely idiosyncratic, not related to any center or representing gradients of expansions (or contractions) from these core areas. 

NOVELTY OF THE METHOD PROPOSED

In the abstract (line 39), it is said that the SCAN “approach eliminates or reduces limitations of other methods and permits pattern description without hidden assumptions about processes, and so should make a valuable contribution to the biogeographer’s toolbox”. The value of this contribution should be regarded to the methods already available in that toolbox, and here the authors should make a clarification effort.

Response: We agree with the commentary and added references and comments pertaining this subject in the new version of the text. Particularly, the clear distinction between the chorological and regionalization approaches contributed to clarify this aspect, and the relation between SCAN and other methods. The fuzzy approach was cited in the introduction, as well the endemicity and bioregion methods, as alternative to ‘standard’ protocols with special capabilities to solve common problems in detecting patterns of species shared ranges.

Congruence vs. similarity:

Mathematically, the “spatial congruence index” (line 135) is the equivalent, in the polygon approach, to Jaccard’s index in the grid approach. Congruence and similarity are, so, synonymous concepts in the search for biotic elements. 

Response: Indeed, the indexes are similar, but not identical. A generic Jaccard {(A ⋂ B) / (A ⋃ B)} gives distinct results from the Cs index, which is more ‘conservative’ (e.g., two 2x2 squares with 50% overlap (Fig 1A) have a similarity of 1/3 for Jaccard and ¼ for Cs). Anyway, the choice of the better index is still an open subject, to be tested opportunistically, and any quantitative index may be used by SCAN to detect congruence.

For decades, similarity has been used in the analysis of biotic elements, under the consideration of its deep biogeographical meaning [e.g. Baroni-Urbani & Collingwood (1977) Acta Zoologica Fennica 152: 1-34]. So, I think that the sentence “degree of congruence (congruence threshold) is an intuitively simple concept LIKELY to have biological relevance” (line 421) should be reconsidered.

Response: We agree and use the term generic (line 133 of the original paper) for the index. The sentence was fixed excluding the word ‘likely’.

Objectivity:

The SCAN procedure is based on the definition of different thresholds (see lines 194 to 199), which values are “preliminarily defined after pilot tests” (line 318). In fact, the authors recognize that “there is no theoretical basis for establishing any specific numerical threshold” (line 423). Although it is immediately said that “the method permits exploring alternatives”, I find here a strong drawback in terms of objectivity. Although it is said in the introduction that the “recognition of congruent distributions has traditionally been a relatively subjective process” (line 60), I see that the SCAN is not free from this fault. Grid approaches, instead, able to deal with theoretical concepts such as critical values for the significance of a similarity index [e.g. Baroni-Urbani y Buser (1976) Systematic Zoology 25: 251-259], have provided methods for the objective detection of biotic elements and chorotypes, while preserving the possibility of gradual patterns overlapping with those chorotypes [e.g. Real et al. (1997) African Journal of Ecology 35: 312-325; Real et al. (2008) Global Ecology and Biogeography 17: 735-746].

Response: We understand and agree with the concerns presented by rev #2. And we removed the word “objective” from the title. In our view, however, some confusion about distinct stages of decision can be further clarified. There are two important decisions related to congruence that have to be made, in distinct stages. The first is the a priori setting of parameters, such as the maximum-depth (as rev#2 cited in line 194, item 1). Other settings are of less concern, as they affect the “resolution” of congruence values (item 2), or the minimum congruence value analyzed (item 3). In ‘default’ (relaxed) values (see below), both parameters have little interference over the results. For example, resolution works well at the default 0.01 (1%). Lower resolutions (e.g., 0.5 Ct intervals), although improving computational speeds, lack the precise Cs values in which closed groups change in composition – see Figs 3 and 4, but did not prevent the recognition of these at lower levels, unless the algorithm stops in this 0.5 interval. The minimum Cs, when set to low values (0.1 is the default) is enough to capture most meaningful patterns (the vast majority of species does not have closed groups at this extreme of low congruences – usually the algorithm stops before reaching these low thresholds). As shown in the simulation example, the max-depth limits the amount of indirect connections, which is determined by the study objectives: if desired patterns are cohesive centers of dispersal (e.g., areas of endemism), lower depths are enough, and potential indirect relationships are discarded a priori. For the detection of gradients, relaxed max-depth settings are often required. These parameters can be seen as the “frame” (max Ct, min Ct threshold values in the Y axis, and max-depth in X axis), and the “resolution” (interval between Ct evaluated – Y axis; see Fig 4) in which the natural relationships will be identified. As an analogy, because of the criterion of closed groups, these natural relationships will always be recovered as they are, as a ‘fixed’ background in which the ‘frame’ moves and focuses specific aspects. With settings of max and min Ct adjusted to 1 and 0 (the extreme max and min potential values), and relaxed max depth values (max-depth > 6 or 7, for example), SCAN will recover most, if not all, possible spatial relationships (closed groups) derived from a given reference species. The most important is that the parameters do not change the patterns, as is probably the case for many other methods. They only determine if some of them are recognized or discarded.

The second aspect is the final choice of the appropriate congruence level for presentation of a pattern, a posteriori, which is a more subjective choice in some degree. In fact, we consider it as one of the main ‘virtues’ of the method, because it allows a personalized and contextualized interpretation of the results for each pattern, separately. The whole chorotype is only one, but some partial chorotypes can be chosen as more representative in specific cases (based on other criteria, such as taxonomic, spatial, environmental factors). But the analysis SCAN performs, as demonstrated above, is not subjective at all. If the settings are not restricted by any specific objective of the study, SCAN gives all the possible patterns derived from a species (or a group of them, in case there are synonyms) that can be expressed spatially, in a range of congruence values, and a range of resulting depths. These ranges of congruences and respective depths are important results, which can be interpreted more easily in biological terms. They show how patterns (chorotypes) vary in extent and composition, according to threshold requirements. This type of result is rarely found in other methods. Moreover, this variation can be used to explore trade-offs and distinct sources of spatial influence over the chorotypes, which are highlighted at distinct congruence thresholds. “Endemicity Analysis” of Szumik et al. 2002 can make similar comparisons but it does not have indirect relationships (depth) assessment (by definition cannot detect long gradients extending beyond the tested area). Within SCAN this final ‘personalized’ congruence threshold, if needed, can be chosen among all possibilities expressed as partial chorotypes, as shown by the Figs 3 and 4.

A general criterion, such as the significance value for an index of similarity as used in a cluster-based grid system, can also be applied to SCAN, as the similarity values for partial chorotypes could be compared among distinct closed lists. The consideration of distinct thresholds in distinct contexts, case by case for each reference, is a natural approach for the diversity of syndromes of patterns that can be found in natural communities. In our Amazonian analysis (manuscript in prep.), terrestrial patterns tend to have a ‘core’ center, which is extended through some gradients by additional species incorporated at lower congruence thresholds. River specialists, mostly driven by ecological filters, conversely, have chorotypes characterized by lower levels of congruence or many indirect links, in comparison. 

Overlap, nestedness and relationships between different patterns:

That “partial spatial overlaps may also occur between independent elements” (line 230) is not new in biogeography. The difference between the analysis of biotic elements and the biogeographic regionalization is that, as the authors explain (line 427) the former “allows the identification of patterns overlapped in space but with distinct species compositions”. This is neither new [see, for example, Birks (1976) New Phytology 77: 257-287; or Baroni-Urbani et al. (1978) Memorie della Società Entomologica Italiana 56: 35-92]. In fact, what is qualified as “the most important take-home message of this paper” by the authors (line 502), that is, “the generic use of bioregionalization methods for spatial classification, recognition of biogeographical patterns, and assessment of their historical and ecological drivers”, is neither a novelty. The combined analysis of regionalization and biotic elements whose drivers are explored a posteriori is a quite explored field [e.g. Birks (1976) New Phytology 77: 257-287; Myklestad & Birks (1993) Journal of Biogeography 20: 1-32; Olivero et al. (2013) Systematic Biology 62: 1-21].

Response: Very interesting point. Following rev #1’s suggestions, we explicitly distinguish between these distinct approaches in the paper in more detail, and describing SCAN as a method following the ‘chorological’ approach. Indeed, rev #1 suggested the exclusion of many paragraphs dedicated to regionalization aspects, which we did. This specific paragraph was deleted.

Finally, the SCAN provides other capacities that are of very high interest in biogeography, and this is a remarkable reason for going on with this methodological proposal; but, again, these are not new. One of them is the perception of a link between pattern complexity and causal factor: “highly variable complexes, for instance, congregating from small highly congruent elements to large spatial gradients recovered under relaxed thresholds, may carry information about both vicariant causes, and processes responsible for pattern deconstruction, such as local extinctions, dispersal events, and differential responses controlled by species-specific traits” (lines 442 to 446). Olivero et al. [(2011) Systematic Biology 60: 645-660] and Ferro et al. [(2017) Journal of Biogeography 44: 2145-2160] found, in the high “fuzzy entropy” (i.e. the degree of fuzzyness) of some chorotypes, signs of biogeographic complexity indicating a possible combination of dispersal patterns driven by idiosyncratic responses to ecological factors; whereas a low entropy might indicate a stronger role of history in the chorotype configuration.

Response: We are grateful for these valuable insights. We readily accepted the suggestion and appended this discussion in the same paragraph in the corrected text.

Another one is in the connection vs. disconnection paradox, “the essence of SCAN: regardless of the congruence threshold, if there are closed groups, then there is spatial cohesion among their constituent taxa relative to the pool” (line 483). This resembles the fact that similarity (as congruence) is a fuzzy concept, and so a similarity value higher than 0 already involves a certain degree of similarity. The consequence of this is that any little overlap between two chorotypes already implies that the species forming the biotic element of one are, to a certain degree, members in the other. This fuzziness in the chorotypes limits may be analysed and can drive to interesting hypotheses on their causes and mutual relations [see Olivero et al. (2011) Systematic Biology 60: 645-660]. However, even in the complexity of biogeographic patterns, the basic units in which the fuzziness is found (i.e. biotic elements and chorotypes) should be delimited according to objectively defined thresholds.

Response: This paradox is really intriguing, and poses an extra challenge to the proper understanding of the rationale behind SCAN. We hope this ‘closed list’ criterion under distinct levels of congruence requirements can be appropriately tested and, eventually, proven as a useful concept in chorological biogeography.

---

## [Decision Letter · Decision Letter 1]

27 Apr 2021

PONE-D-21-00528R1

Spatial Congruence Analysis (SCAN): A method for detecting biogeographical patterns based on species’ range congruences

PLOS ONE

Dear Dr. Gatto,

I know received the final reviews of your ms. As you can see, both referees agree (and I also agree too) that you made a strong effort in trying to cover all the issues raised for this paper. Therefore, I kindly ask you to incorporate the very minor changes requested by one of the referees. 

When submitting the revised version of your manuscript, please state in your cover letter point-by-point which changes you have made in response to the referee and where and/or why you have refused to follow a particular suggestion. If the accordance between the changes and the reviewers' requests is sufficiently transparent, no further reviewing will be needed and your ms will be accepted with no further delay. I therefore suggest the author to send their final copy as soon as possible.

We look forward to receiving your revised manuscript.

Kind regards,

Bruno Bellisario, PhD

Academic Editor

PLOS ONE

Journal Requirements:

Reviewers' comments:

Reviewer's Responses to Questions

**Comments to the Author**

1. If the authors have adequately addressed your comments raised in a previous round of review and you feel that this manuscript is now acceptable for publication, you may indicate that here to bypass the “Comments to the Author” section, enter your conflict of interest statement in the “Confidential to Editor” section, and submit your "Accept" recommendation.

Reviewer #1: (No Response)

Reviewer #2: All comments have been addressed

2. Is the manuscript technically sound, and do the data support the conclusions?

Reviewer #1: Yes

Reviewer #2: Yes

3. Has the statistical analysis been performed appropriately and rigorously? 

Reviewer #1: Yes

Reviewer #2: Yes

4. Have the authors made all data underlying the findings in their manuscript fully available?

Reviewer #1: Yes

Reviewer #2: Yes

5. Is the manuscript presented in an intelligible fashion and written in standard English?

Reviewer #1: Yes

Reviewer #2: Yes

6. Review Comments to the Author

Reviewer #1: I congratulate the authors for their excellent revision of the manuscript. I have greatly appreciated their effort to integrate all comments by both referees.

As regards my own, the authors have included virtually all my suggestions and corrections. So, I do not have further major comments.

Minor comments

- I suggest to not use the word “community” (which is typically used at smaller scales in ecological contexts), but the more appropriate words “floras” and “faunas” (which refer to larger species assemblages). For example: line 25 “in a community” -> “in a flora or fauna”; line 445: communities -> Floras and faunas

- Line 75: The “use of indexes”. Do you mean similarity coefficients? In such case, they are part of clustering techniques. I think that the sentence might be simpler, clearer and more comprehensive as follows: “The use of clustering and grid-based bipartite network approaches etc.”

- 77: information-bearing unit -> information-bearing units

- 79: “may be biased by the use of standard cluster-based methods” -> “may produce contrasting results according to specific clustering method used”

- 87: allow -> allows

- 92-93: Delete “Areas identified can be seen as analogous to chorotypes”. It is unnecessary and creates confusion.

- 102-103: Delete “that have overlapping distributions”. It is unnecessary and creates confusion.

- 108. Delete “We show that SCAN can solve specific biogeographical problem cases as other methods recently proposed”. It is unclear and not useful.

- 109-112. I suggest rephrasing as follows: “SCAN’s mathematically simple and intuitive approach of direct range map comparisons offers an alternative to other available methods, using different computational or conceptual bases, and provides new parameters and metrics that can be useful in interpreting biogeographical patterns”.

- 202: Please, explain which these default setting values are (only for CT a value is given, 0.1). Also, try to provide some justification (theoretical or empirical) for the proposed default setting values.

- 228: “constitutes the chorotype” -> constitutes a chorotype

- 325. bird community -> avifauna

- 449 delete space after [58]

- 450 delete space after [37]

- For consistency, change “biogeographic” to “biogeographical” and “chorologic” to “chorological” in all instances

- Although the author removed any reference to “biotic element”, this expression is still present at places, and the word “element” is used in an unclear way: see lines 296, 300-301, 416, 475, 665, 666, 669, 717

- Finally, appreciate the way fuzzy approaches are discussed, as they are uncommonly used and cited. I wonder if the use of medoids to identify representative species (Fattorini, S. 2007, A statistical method for idiographic analyses in biogeographical research. Diversity and Distributions, 13: 836-844. https://doi.org/10.1111/j.1472-4642.2007.00400.x) might represent a fuzzy approach connecting the “chorological” and “regionalization” programmes.

Reviewer #2: In their answers to the referees, the authors have made a deep discussion of the points I presented in my review. I have seen a strong effort in the new manuscript in order to address my main concern, consisting on the need to discuss the contribution of SCAN in the light of other available approaches. Also, the use of "chorotype" as the biogeographic unit to which the analysis output is referred makes the paper easier to understand and compare with the previous literature. Now I think that this manuscript is worth publishing in its present state.

7. PLOS authors have the option to publish the peer review history of their article (what does this mean?). If published, this will include your full peer review and any attached files.

Reviewer #1: No

Reviewer #2: No

---

## [Author Response · Author response to Decision Letter 1]

3 May 2021

Manaus, 02 May 2021

Dear Academic Editor Bruno Bellisario:

We are very pleased with the results of the revision process - the paper was greatly improved by the suggestions and comments provided by the reviewers. In this last round, we tried to follow accurately the suggestions of the first reviewer. In addition, we made some small modifications of our own to improve clarity in a few specific sentences, with no major content modifications. The points raised by the reviewer are listed below and commented regarding their new configuration in the updated manuscript. All changes can be tracked in its commented version.

This whole revision and correction process led to a significant improvement in the manuscript. For this, we are especially grateful to both reviewers. Although we noted that they chose to remain anonymous, we are very eager to thank them publicly. We wonder if you might ask them to reconsider revealing their names so we might acknowledge their contributions specifically in the acknowledgments. After these modifications, the manuscript is ready for acceptance. Please let us if there is any pending problem. By the rules of our graduate program, CG requires formal acceptance of the manuscript to schedule his Ph. D. defense.

Thank you again for your valuable insights and kind conduction of the whole process.

With warm regards,

Cassiano Gatto & Mario Cohn-Haft

Comments and responses:

Reviewer #1: I congratulate the authors for their excellent revision of the manuscript. I have greatly appreciated their effort to integrate all comments by both referees. As regards my own, the authors have included virtually all my suggestions and corrections. So, I do not have further major comments.

Minor comments

- I suggest to not use the word “community” (which is typically used at smaller scales in ecological contexts), but the more appropriate words “floras” and “faunas” (which refer to larger species assemblages). For example: line 25 “in a community” -> “in a flora or fauna”; line 445: communities -> Floras and faunas

Response: lines 23, 170, 274, 295, 297, 322, 366, 417, 427, 429, 443, 651, 671, 672 were modified. The 'community' term was deleted or replaced by other context-dependent terms in each case.

- Line 75: The “use of indexes”. Do you mean similarity coefficients? In such case, they are part of clustering techniques. I think that the sentence might be simpler, clearer and more comprehensive as follows: “The use of clustering and grid-based bipartite network approaches etc.”

Response: we followed the suggestion

- 77: information-bearing unit -> information-bearing units

Response: fixed

- 79: “may be biased by the use of standard cluster-based methods” -> “may produce contrasting results according to specific clustering method used”

Response: we followed the suggestion

- 87: allow -> allows

Response: Fixed

- 92-93: Delete “Areas identified can be seen as analogous to chorotypes”. It is unnecessary and creates confusion.

Response: Deleted

- 102-103: Delete “that have overlapping distributions”. It is unnecessary and creates confusion.

Response: Deleted

- 108. Delete “We show that SCAN can solve specific biogeographical problem cases as other methods recently proposed”. It is unclear and not useful

Response: Deleted

- 109-112. I suggest rephrasing as follows: “SCAN’s mathematically simple and intuitive approach of direct range map comparisons offers an alternative to other available methods, using different computational or conceptual bases, and provides new parameters and metrics that can be useful in interpreting biogeographical patterns”.

Response: Rephrased as suggested

- 202: Please, explain which these default setting values are (only for CT a value is given, 0.1). Also, try to provide some justification (theoretical or empirical) for the proposed default setting values.

Response: We expanded this paragraph to accommodate a more detailed presentation of these parameters, and their default values, and a proper justification (see 195 Control parameters).

- 228: “constitutes the chorotype” -> constitutes a chorotype

Response: Fixed

- 325. bird community -> avifauna

Response: rephrased

- 449 delete space after [58]; 450 delete space after [37]

Resp: Deleted

- For consistency, change “biogeographic” to “biogeographical” and “chorologic” to “chorological” in all instances

Resp: Replaced in all instances

- Although the author removed any reference to “biotic element”, this expression is still present at places, and the word “element” is used in an unclear way: see lines 296, 300-301, 416, 475, 665, 666, 669, 717

Response: Occurrences of the term element were replaced by equivalent terms, such as pattern, or chorotypes, according to the diverse contexts in which it appeared.

- Finally, appreciate the way fuzzy approaches are discussed, as they are uncommonly used and cited. I wonder if the use of medoids to identify representative species (Fattorini, S. 2007, A statistical method for idiographic analyses in biogeographical research. Diversity and Distributions, 13: 836-844. https://doi.org/10.1111/j.1472-4642.2007.00400.x) might represent a fuzzy approach connecting the “chorological” and “regionalization” programmes.

Response: Both analyses compare the distances intra and inter cluster. While the medoid approach chooses the most ‘representative’ species, the fuzzy approach calculates how much these clusters have species with multiple potential membership. But their background ideas are similar, comparisons between clusters. The identification of representative species in SCAN follows another path. It only chooses the species with highest average congruence to the other component species of the chorotype.

Journal Requirements:

Please review your reference list to ensure that it is complete and correct. […]

Response: Following the journal requirements we updated the references fixing some minor problems, such as misspelled author names and extra spaces. 262 – reference number updated; 276 – a more appropriate paper replaced the former reference. The section References was updated and all references manually checked to ensure correct spelling and proper citation of papers.

---

## [Editor Report · Decision Letter 2]

10 May 2021

Spatial Congruence Analysis (SCAN): A method for detecting biogeographical patterns based on species’ range congruences

PONE-D-21-00528R2

Dear Dr. Gatto,

We’re pleased to inform you that your manuscript has been judged scientifically suitable for publication and will be formally accepted for publication once it meets all outstanding technical requirements.

Kind regards,

Bruno Bellisario, PhD

Academic Editor

PLOS ONE

---

## [Editor Report · Acceptance letter]

12 May 2021

PONE-D-21-00528R2 

Spatial Congruence Analysis (SCAN): A method for detecting biogeographical patterns based on species range congruences 

Dear Dr. Gatto:

I'm pleased to inform you that your manuscript has been deemed suitable for publication in PLOS ONE. Congratulations! Your manuscript is now with our production department. 

Kind regards, 

on behalf of

Dr. Bruno Bellisario 

Academic Editor

PLOS ONE